# Integrated 2D multi-fin field-effect transistors

Mengshi Yu[1,4], Congwei Tan[1,4], Yuling Yin[2,3,4], Junchuan Tang[1], Xiaoyin Gao[1], Hongtao Liu[1], Feng Ding [2,3] ✉ & Hailin Peng [1] ✉

Vertical semiconducting fins integrated with high-$\kappa$ oxide dielectrics have been at the centre of the key device architecture that has promoted advanced transistor scaling during the last decades. Single-fin channels based on two-dimensional (2D) semiconductors are expected to offer unique advantages in achieving sub-1 nm fin-width and atomically flat interfaces, resulting in superior performance and potentially high-density integration. However, multi-fin structures integrated with high-$\kappa$ dielectrics are commonly required to achieve higher electrical performance and integration density. Here we report a ledge-guided epitaxy strategy for growing high-density, mono-oriented 2D $Bi_2O_2Se$ fin arrays that can be used to fabricate integrated 2D multi-fin field-effect transistors. Aligned substrate steps enabled precise control of both nucleation sites and orientation of 2D fin arrays. Multi-channel 2D fin field-effect transistors based on epitaxially integrated 2D $Bi_2O_2Se/Bi_2SeO_5$ fin-oxide heterostructures were fabricated, exhibiting an on/off current ratio greater than $10^6$, high on-state current, low off-state current, and high durability. 2D multi-fin channel arrays integrated with high-$\kappa$ oxide dielectrics offer a strategy to improve the device performance and integration density in ultrascaled 2D electronics.

Over the past few decades, conventional field-effect transistors (FETs) based on three-dimensional (3D) semiconductors have continued to shrink in size according to Moore's Law (the number of transistors in an integrated circuit doubles approximately every two years), enabling improvements in device performance and transistor density[1,2]. To sustain further increases in integration density and improvements in electrical performance, the fin field-effect transistor (FinFET) featuring a thin vertical fin channel surrounded by the gate electrodes was adopted at the 22 nm node[3]. This nonplanar transistor structure provides superior electrostatic control and marked reduction in energy consumption[4–7], enabling the continuous downscaling of integrated circuits (ICs)[3,7–14].

However, Si-based FinFETs are still struggling to achieve sub-3 nm nodes with minimum fin width of about 5 nm that reaches the physical limitation presented by quantum mechanical issues. In particular, the short-channel effect and sub-5 nm channel thickness ultimately limit Si-based FinFET downscaling. The reduced mobility, imperfect interfaces, and nonuniform electrostatic control associated with channel surface and fin shape are the main obstacles. On the one hand, the worsened surface roughness of thinning 3D channel material induces strong surface scattering of charge carriers[15,16]. Additionally, the tapered fin shape of Si FinFETs, which presumably originated from the top-down etching process of bulk Si[17,18], leads to poorer electrostatic control at the fin bottom with larger width[17–19] (Fig. 1a). In order to overcome these issues, efforts have been devoted to fabricate 3D transistor architectures using 2D semiconductors[20–27], especially vertical 2D FinFETs[20–23], which can be attributed to the unique advantages of 2D material with atomically flat surface[28–33] and the rectangular fin

[1]Center for Nanochemistry, Beijing Science and Engineering Center for Nanocarbons, Beijing National Laboratory for Molecular Sciences, College of Chemistry and Molecular Engineering, Peking University, Beijing, China. [2]Institute of Technology for Carbon Neutrality, Shenzhen Institute of Advanced Technology, Chinese Academy of Sciences, Shenzhen, China. [3]Faculty of Materials Science and Energy Engineering, Shenzhen University of Advanced Technology, Shenzhen, China. [4]These authors contributed equally: Mengshi Yu, Congwei Tan, Yuling Yin. ✉e-mail: f.ding@siat.ac.cn; hlpeng@pku.edu.cn

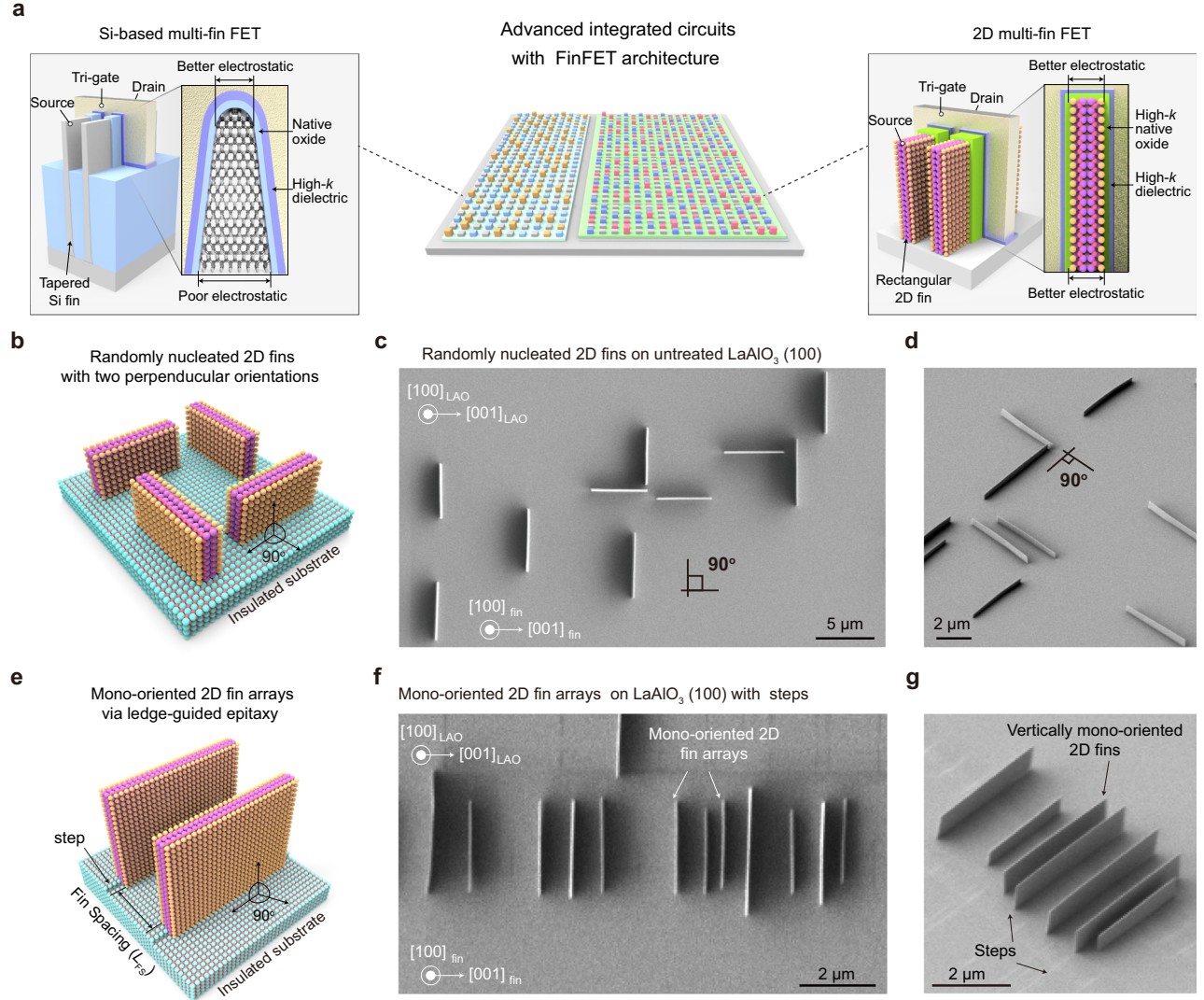

**Fig. 1 | Ledge-guided epitaxy of mono-oriented vertical 2D fin arrays for 2D multi-fin field-effect transistors (FETs). a** Schematic illustration for 2D multi-fin FETs potentially applied in advanced integrated circuits. **b**–**d** Schematic (**b**) and corresponding scanning electron microscopy (SEM) images of 2D $Bi_2O_2Se$ fins with two perpendicular orientations on pristine $LaAlO_3$ (100) surface, including top view (**c**) and tilted view (**d**). **e**–**g** Schematic (**e**) and corresponding SEM images of ledge-guided epitaxial mono-oriented 2D $Bi_2O_2Se$ fins on pre-treated $LaAlO_3$ (100) surface, including top view (**f**) and tilted view (**g**).

shape originated from bottom-up growth method[20]. Remarkably, 2D $Bi_2O_2Se/Bi_2SeO_5$ fin/oxide heterostructure enables 2D single-fin FETs integrated with high-$\kappa$ gate oxide, exhibiting comparable performance to industrial Si-based FinFETs[23].

In principle, the ultrascaled FinFETs tend to adopt uniform multi-fin arrays to achieve high-density integration and robust performance in terms of power gain and drive capability for logic chips[34–37]. As compared to single-fin FETs, multi-fin FETs might offer a larger total channel width, resulting in higher total drive current, transconductance, and lower noise[35]. Despite the advanced performance of 2D single-fin FETs, 2D multi-fin FETs are also highly desired for enhanced power gain and drive capability. Therefore, it is necessary to explore the growth of high-density 2D fin arrays for the fabrication of integrated 2D multi-fin FETs.

Here, we present a ledge-guided epitaxy approach to grow high-density, mono-oriented 2D fin arrays on diverse insulating substrates for the fabrication of integrated 2D multi-fin FETs. Remarkably, the ledge-guided epitaxy of mono-oriented 2D fin arrays is independent of the symmetry of substrate and differs to the recently reported defect-induced epitaxy which necessitates coworkers with two-fold symmetric substrates (e.g., MgO (110)) and defect-induced selective

nucleation[23]. The pre-created aligned steps on the substrates assist in controlling the nucleation sites and orientation of 2D $Bi_2O_2Se$ fin arrays via reducing the binding energy of nuclei at the step edge and lowering the substrate symmetry, enabling the growth of high-density parallel 2D fin arrays with a minimum fin spacing of sub-20 nm. As-grown 2D $Bi_2O_2Se$ fin arrays were fabricated into multi-channel 2D FinFETs with integrated high-$\kappa$ native-oxide $Bi_2SeO_5$, which exhibit superior electrical performance and good durability, including low off-state current ($I_{OFF}$), large on/off current ratio ($I_{ON}/I_{OFF}$) of >$10^6$, and high on-state current ($I_{ON}$). Compared with 2D single-fin FETs, 2D multi-fin FETs possess superior electrical performance, such as enhanced on-state current and transconductance, indicating the potential of high-density 2D fin-oxide heterostructure arrays for realizing ICs with high-density integration and improved performance.

## Results

### Ledge-guided epitaxy of 2D fin arrays

As an emerging 2D semiconductor, $Bi_2O_2Se$ has unique layered structure and superior properties, including ultrahigh carrier mobility, moderate band gap, great air stability[38,39]. Notably, $Bi_2O_2Se$ has a high-$\kappa$ ($\kappa \approx 21$) competitive native-oxide dielectric, $Bi_2SeO_5$,

which works as effectively as $SiO_2$ does with $Si^{40-42}$. $Bi_2O_2Se$ has a tetragonal crystal structure ($I4/mmm$, $a = 3.88$ Å, $c = 12.16$ Å and $Z = 2$), and consists of positively charged $[Bi_2O_2]_n^{2n+}$ layers sandwiched by negatively charged $Se_n^{2n-}$ layers (Supplementary Fig. 1). There are relatively weak electrostatic interactions between the $[Bi_2O_2]_n^{2n+}$ layers and the $Se_n^{2n-}$ layers, while within the $[Bi_2O_2]_n^{2n+}$ layers are strong covalent bonds. The high density of dangling bonds at an edge of layered $[Bi_2O_2]_n^{2n+}$ greatly enhances the bonding between the edge and the substrate and allows vertical 2D $Bi_2O_2Se$ fins to be efficiently grown on diverse insulating substrates (such as $LaAlO_3$, MgO and $CaF_2$). On the other hand, the in-plane orientation of 2D fins depends on the symmetry matching between the substrate and 2D fin. On pristine $[LaO]^+$-terminated $LaAlO_3$ (100) surface with 4-fold symmetry (Fig. 1 b–d), the 2-fold symmetric 2D $Bi_2O_2Se$ fins have two equivalent energy-minimum orientations (that is, 0° and 90° orientations) and were found randomly aligned along two perpendicular directions of the substrate. Due to the two different in-plane orientations and the nature of random nucleation, the as-grown 2D fins were discrete, which impedes further high-density integration of 2D multiple fin channels.

To grow unidirectionally aligned 2D fin arrays, we first prepared high-density, aligned steps on insulating substrates, which had been confirmed effective for the growth of aligned 2D single crystals[43]. Taking the adopted $LaAlO_3$ (100) as a presentative example, several artificially self-aligned steps with specific orientation on the $LaAlO_3$ (100) surface can be easily pre-created with parallel or perpendicular the [001] or [010] direction by using a less-sharp diamond scraper before the growth of 2D fin arrays (Supplementary Fig. 2). Remarkably, the self-aligned step edges formed with an "atomic" resolution originate from brittle fractures along the [001] or [010] direction of the $LaAlO_3$ lattice from the (010) or (001) cleavage plane. From the experimental epitaxial results, despite the fact that the formed step edges are not perpendicular to the scratches, a mono-oriented 2D fin array is still obtained (Supplementary Fig. 2). With the assistance of these aligned steps, the vertical 2D $Bi_2O_2Se$ nucleus are anchored along the pre-patterned step edges to site-specifically grow unidirectionally aligned 2D fin arrays (Fig. 1e–g and Supplementary Fig. 3). The whole ledge-guided epitaxy towards the energetically favorable growth of aligned 2D fin arrays mainly involves the following four processes (taking $LaAlO_3$ as a representative example): (i) a single-crystal epitaxy substrate with exposed ledges is adopted; (ii) exposed ledges on the substrate surface preferentially trap precursor atoms and thereby serve as nucleation sites; (iii) 2D fin seeds with energetic minimum nucleate at the ledge, breaking the symmetry and selectively stabilizing a preferred orientation; (iv) mono-oriented seeds grow anisotropically into well-aligned 2D fins (Supplementary Fig. 4). Remarkably, after combining a micromachined arm and a diamond scrape, the spacing of 2D fin arrays is controllable by controlling the spacing of step arrays (Supplementary Fig. 2).

### Structure characterization of ledge-guided epitaxial 2D fins

To elucidate the interfacial microstructures and nucleation mechanism of vertical 2D $Bi_2O_2Se$ fin arrays grown by the ledge-guided epitaxy, we have performed aberration-corrected scanning transmission electron microscopy (AC-STEM) investigations. The 2D $Bi_2O_2Se$ fin/$LaAlO_3$ slices were initially fabricated using focused ion beam (FIB) milling, followed by extensive cross-sectional AC-STEM observations. Figure 2a shows a typical 2D $Bi_2O_2Se$ fin grown by ledge-guided epitaxy, which is strictly perpendicular to the $LaAlO_3$ substrate surface with a smooth surface along its fin height. The aspect ratio (i.e., height/thickness) of this particular 2D fin is as high as ~37 (~30 nm in thickness, ~1.1 μm in height), which may help boost the electrical performance of 2D FinFETs[44]. High-resolution (HR) AC-STEM image of the vertical 2D $Bi_2O_2Se$ fin from the side view (Fig. 2b) indicates the layered structure

of 2D $Bi_2O_2Se$ with a layer spacing of ~0.61 nm, consistent with the (002) planes of layered $Bi_2O_2Se$. The corresponding Fast Fourier Transform (FFT) diffraction spots of the interface further indicate the epitaxial relationships where the (100) and (001) planes of epitaxial $Bi_2O_2Se$ are parallel to (100) and (001) planes of $LaAlO_3$, respectively (Fig. 2c). Note that atomically sharp steps exist on the $LaAlO_3$ substrate surface (Fig. 2b), allowing for guided growth of vertical 2D $Bi_2O_2Se$ fin. Interestingly, the interface strain of 2D fin around the substrate step is clearly identified (Fig. 2d), which can be almost completely relaxed along the vertical direction of the 2D fin within ~1.0 nm. Furthermore, the enlarged image of interface microstructure as shown in Fig. 2e exhibits the atomically sharp interface and perfect epitaxial growth between $Bi_2O_2Se$ fin and the step edge of $[LaO]^+$ terminated $LaAlO_3$ (100) surface.

### Ledge-guided epitaxy mechanism of 2D fin arrays

Density functional theory (DFT) calculations were performed to gain a better understanding of the ledge-guided epitaxy mechanism. As shown in Fig. 2f, g, the binding energy of a Bi/Se atom or Bi-O monomer at the step edge and terrace of $[LaO]^+$ terminated $LaAlO_3$ substrate surface was calculated, respectively. The results showed that the binding energy of Bi atom at the terrace is 0.48 eV, while it decreases to $-0.26$ eV at the step edge. For Se atom adsorption, the binding energy is $-0.45$ eV at the terrace and it decreases to $-1.06$ eV at the step edge. Supplementary Fig. 5 illustrates additional adsorption structures of Bi and Se, which further confirm that Bi and Se atoms tend to preferentially absorb at the step edge site rather than the terrace site. Similarly, the Bi–O monomer has lower binding energy at the step edge than terrace, indicating that the exposed step edges preferentially trap precursor Bi–O monomers (Fig. 2g).

We further optimized the structure and calculated the binding energy of the 2D $Bi_2O_2Se$ nucleus at the step edge and the terrace of $LaAlO_3$, respectively (Fig. 2g). The calculations clearly reveal that the nucleation of vertical 2D $Bi_2O_2Se$ fins at the step edge of $LaAlO_3$ substrate is highly preferred than that on terrace. The 2D $Bi_2O_2Se$ fin nucleus exhibit a lower binding energy of $-10.40$ eV at the step edge than that on the terrace, $-9.62$ eV. The energy difference ($\Delta E$) between two nucleation sites reach 0.78 eV and thermodynamic probability of the 2D $Bi_2O_2Se$ fin nucleation at the step edge can be roughly estimated to be $1/(1 + \exp(\Delta E/k_BT)) > 99.99\%$, in which $k_B$ is the Boltzmann constant and $T = 903$ K is the growth temperature of 2D fin arrays. Considering that the step edges on the substrate serve as active lines to initiate the nucleation of 2D $Bi_2O_2Se$ fins, the site-specific epitaxy of 2D fin arrays can be well controlled via the ledge-guided epitaxy.

### Orientation and density control of 2D fin arrays

Besides controlling the nucleation sites, the aligned step edges of the substrate also enable the mono-orientation alignment of 2D fins on the substrate. The 2-fold symmetric 2D $Bi_2O_2Se$ fins vertically grown on a 4-fold symmetric $LaAlO_3$ substrate have two energy-degenerated orientations, which are perpendicular to each other. However, by introducing high-density aligned steps on the $LaAlO_3$ substrates, the substrate symmetry was reduced from $C_{4V}$ to $C_{2V}$. According to the principle of symmetry matching[45-47], the nucleated $C_{2V}$ symmetric $Bi_2O_2Se$ fins can be mono-oriented on a pretreated $C_2$ symmetric $LaAlO_3$ substrate, resulting in the vertical growth of aligned 2D fin arrays at the controlled nucleation sites.

In addition to artificial step edges, reducing the substrate symmetry by applying a miscut angle can also be used to tune the in-plane orientation of 2D $Bi_2O_2Se$ fin arrays (Fig. 3a). Miscut toward [100] produces nanoscale steps on $LaAlO_3$ (100) surface with their edges along [010]. On well-cut $LaAlO_3$ (100) surface with a tunable miscut angle and step density, the orientation of vertical 2D $Bi_2O_2Se$ fin arrays can be adjusted from two perpendicular orientations to mono-orientation as the miscut angle toward the [100] direction increases

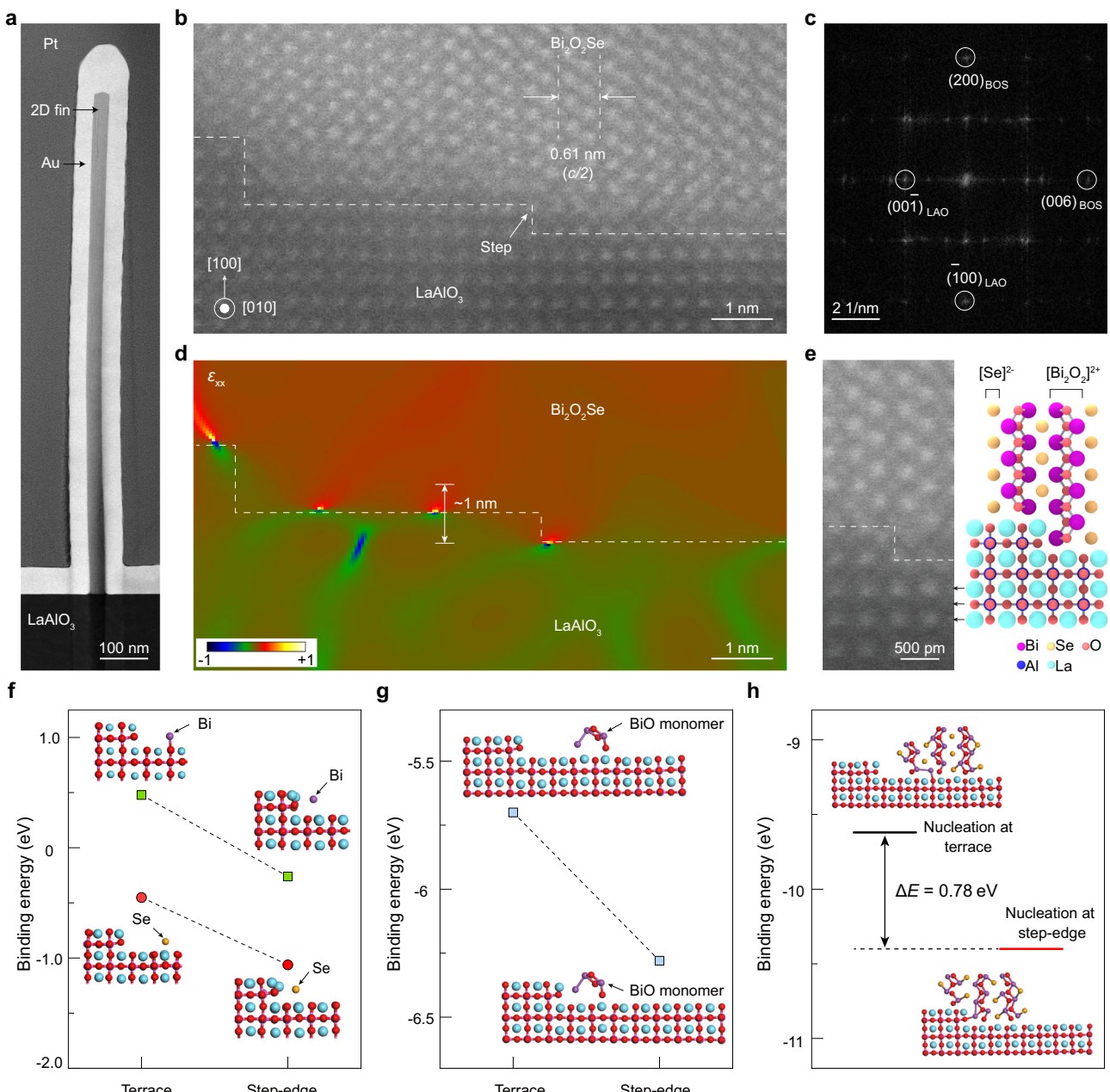

**Fig. 2 | Structure characterization and nucleation mechanism of vertical 2D fin grown by ledge-guided epitaxy. a** Cross-sectional low-magnification transmission electron microscopy (TEM) image of a vertical $Bi_2O_2Se$ fin grown by ledge-guided epitaxy. **b** Cross-sectional high-angle annular dark-field scanning transmission electron microscopy (HAADF-STEM) image showing clear steps on the surface of the substrate, where the fin nucleated. The dashed line means $Bi_2O_2Se$/LaAlO$_3$ interface. **c** Fast Fourier Transform (FFT) diffraction spots of (**b**). **d** Strain mapping ($\varepsilon_{xx}$) estimated from a filtered version of the panel (**b**). **e** High-magnification HAADF-STEM image with atomic resolution and corresponding schematic of $Bi_2O_2Se$/LaAlO$_3$ interface. The dashed line represents actual interface. **f, g** The binding energies and optimized structures of Bi/Se atoms (**f**) and Bi-O monomers (**g**) adsorbed at the step edge and the terrace of the LaAlO$_3$ substrate, respectively. **h** The binding energies and optimized structures of a 2D $Bi_2O_2Se$ nucleus at the step edge and terrace, demonstrate that nucleation at the step edge is energetically favorable. $\Delta E$ is the energy difference between two nucleation sites.

from 0.1° to 10° (Fig. 3b). As the miscut angle increases, the correspondingly increased step density inhibits the across-terrace growth of 2D $Bi_2O_2Se$ fins, forming mono-oriented 2D fins. When miscut angle of LaAlO$_3$ (100) substrates reaches ~10°, purely mono-oriented 2D $Bi_2O_2Se$ fin arrays were obtained (Figs. 3a, b). Remarkably, we used high-density aligned steps of LaAlO$_3$ (100) substrate to further increase the density of aligned 2D fins via the ledge-guided epitaxy (Fig. 3c).

This strategy is also applicable for other insulated substrates, such as MgO and CaF$_2$. Unidirectionally aligned 2D $Bi_2O_2Se$ fins nucleate randomly and grow anisotropically on pristine 2-fold symmetric MgO

(110) and CaF$_2$ (110) surfaces (Supplementary Fig. 6). With the introduction of high-density aligned steps on the surface of MgO (110) and CaF$_2$ (110) slices, high-density 2D $Bi_2O_2Se$ fin arrays are achievable (Fig. 3d, e). Figure 3f presents the statistic result of the minimum fin spacing and fin pitch in various 2D fin arrays on different substrates. As-grown 2D fin arrays generally exhibit small fin spacing of less than 80 nm and fin pitch of less than 200 nm (Supplementary Fig. 7), among which some 2D fin arrays even achieve a minimum fin spacing of less than 20 nm (Fig. 3f). Given more precise and controllable preparation of substrate steps, it will be possible to fabricate ordered high-density

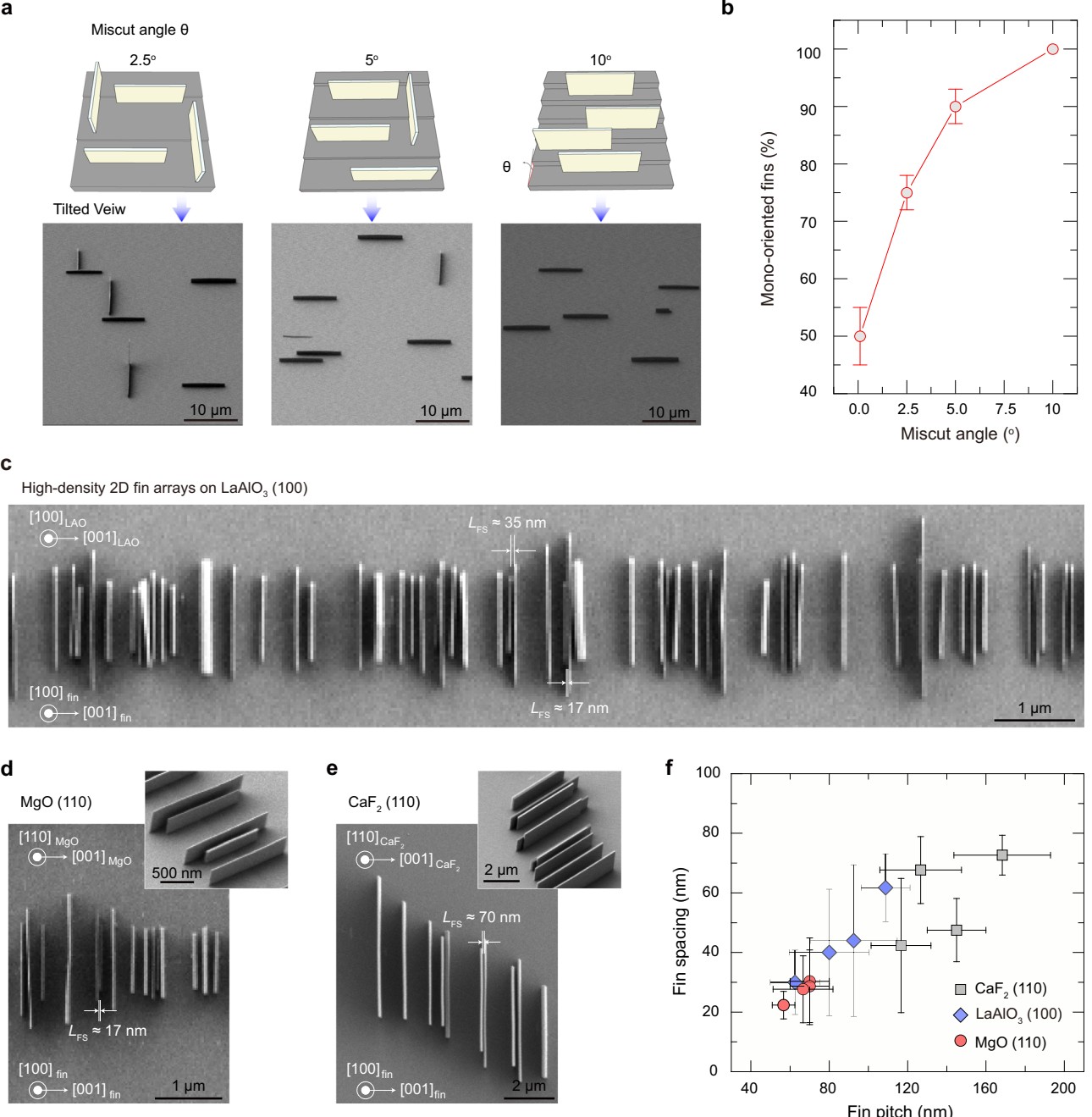

**Fig. 3 | Guided-growth of mono-oriented high-density 2D fin arrays for 2D multi-fin FETs with the assistance of steps. a** Schematic and SEM images showing the effect of the step density on the orientation of 2D $Bi_2O_2Se$ fin arrays. **b** Statistic for mono-oriented fin percentage of 2D $Bi_2O_2Se$ fin arrays growth with miscut angle of the epitaxial substrate. Error bars indicate standard deviations of mono-oriented fin percentage for different miscut angle. **c**–**e** SEM images of high-density 2D

$Bi_2O_2Se$ fin arrays grown by ledge-guided epitaxy on $LaAlO_3$ (100) surface (**c**), MgO (110) surface (**d**) and $CaF_2$ (110) surface (**e**). Insets: corresponding tilted-view high-magnification SEM images. $L_{FS}$ represents fin spacing. **f** Statistical minimum fin pitch and fin spacing of different vertical 2D $Bi_2O_2Se$ fin arrays grown by ledge-guided epitaxy on various substrates. Error bars indicate standard deviations of minimum fin spacing and fin pitch for different 2D fin arrays.

2D fin arrays that meet the material requirement of advanced sub-1 nm technology node, as projected by International Roadmap for Devices and Systems (IRDS)[48].

### Electrical performance of 2D multi-fin FETs

High-density aligned 2D fin arrays facilitate the fabrication of 2D multi-fin FETs that integrate multiple fin channels, which have the potential to boost electrical performance with higher drive capability. As illustrated in Fig. 4a–e, conformal formation of epitaxial native-oxide $Bi_2SeO_5$ layer and atomic layer deposition of $HfO_2$ film are both used as

high-$\kappa$ bilayer dielectrics of 2D multi-fin FETs. Epitaxial $Bi_2SeO_5$ dielectric ($\kappa \approx 21$) was created by intercalation oxidation of 2D $Bi_2O_2Se$ fin with the assistance of ultraviolet, where $Se_n^{2n-}$ layers underwent intercalative oxidation to $SeO_3^{2-}$ groups while the layered $[Bi_2O_2]_n^{2n+}$ framework structures remain intact (Supplementary Fig. 8). In the space between two $[Bi_2O_2]_n^{2n+}$ layers, a $SeO_3^{2-}$ group has four equivalent orientations. Our DFT calculations indicate that the $SeO_3^{2-}$ groups between two $[Bi_2O_2]_n^{2n+}$ layers tend to be aligned along one direction, while tuning the orientations of $SeO_3^{2-}$ groups in neighboring layers only results in very small energy difference and lattice constant

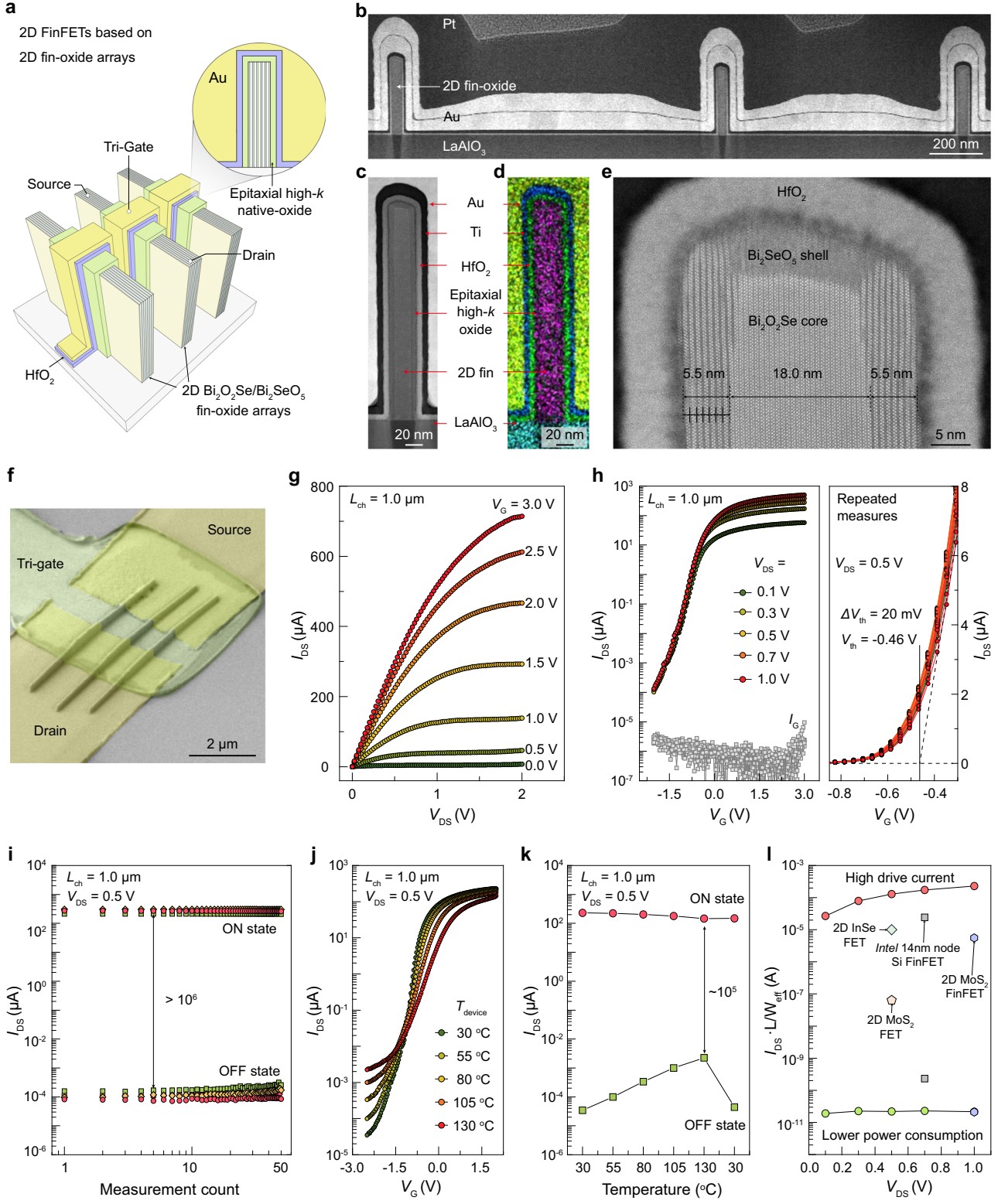

change. For example, the three structures of $Bi_2SeO_5$ shown in Supplementary Fig. 9 have nearly equivalent lattice constants and stability. Notably, the fin-oxide heterostructure with atomically smooth interface, formed by the 2D $Bi_2O_2Se$ fin integrated high-$k$ native-oxide $Bi_2SeO_5$ epilayer, acts as the workhorse architecture for 2D multi-fin FETs (Supplementary Fig. 8d). On the one hand, controllable oxidation is an effective approach to thinning 2D fins in the fin-oxide heterostructures, which enables 2D fin as thin as 3 nm (Supplementary

Fig. 8e). In addition, combining oxidation and selective etching via diluted HF acid (-0.2 %), the downscaling of total thickness of 2D fins can possibly be further realized (Supplementary Fig. 10).

2D multi-fin FET with channel length of about 1 μm was fabricated based on three $Bi_2O_2Se$ fins with relatively consistent shapes (Fig. 4f). The 2D $Bi_2O_2Se$ fin channel is ~250 nm high and 24 nm thick, surrounded by a 5.5 nm-thick epitaxial high-$k$ $Bi_2SeO_5$ and 7.2 nm-thick $HfO_2$ dielectric (Supplementary Fig. 12). The fin spacing of about 1 μm

**Fig. 4 | Electrical performance of 2D multi-fin FinFETs based on aligned 2D Bi$_2$O$_2$Se-Bi$_2$SeO$_5$ fin-oxide arrays. a** Schematic diagram of 2D Bi$_2$O$_2$Se/Bi$_2$SeO$_5$/HfO$_2$ FinFET with three fins. **b** Cross-sectional STEM image of fin arrays. **c–e** Low-magnification STEM image (**c**), Energy-dispersive X-ray spectroscopy (EDS) image (**d**), and high-magnification STEM image (**e**) of Bi$_2$O$_2$Se/Bi$_2$SeO$_5$ fin-oxide heterostructures covered with HfO$_2$ dielectric layer. **f** Tilted SEM image of a 3-fin FinFET. **g** Typical output curves of the 2D multi-fin FET in (**f**). $I_{DS}$ is the source-drain current. $V_{DS}$ is the source-drain voltage and $V_G$ is the gate voltage. $L_{ch}$ represents the channel length of the devices. **h** Transfer curves of the 2D multi-fin FET in (**f**) and the repeated transfer curve measurement results for 50 cycles of the 2D multi-fin FET.

$I_G$ is the gate leakage current, $V_{th}$ is the threshold voltage and $\Delta V_{th}$ means the shift of threshold voltage. The intersection of the two dashed lines is the threshold voltage of the device, represented by a solid line. **i** Statistical on-state current ($I_{ON}$) and off-state current ($I_{OFF}$) measured over 50 cycles for different multiple-channel 2D FinFETs. **j, k** Transfer curves (**j**) and statistical $I_{ON}$ and $I_{OFF}$ (**k**) of the 2D multi-fin FET in (**f**) operated under different temperatures. **l** Comparison of normalized current of the fabricated 2D multi-fin FETs with 2D MoS$_2$ FET[49], 2D InSe FET[50], 2D MoS$_2$ FinFETs[20] and Intel's 14 nm-node Si FinFETs[51] under low gate-voltage modulation. $L$ is the channel length and $W_{eff}$ is the effective width of device.

was chosen for the convenience in device fabrication processing. The typical output and transfer curves are plotted in Fig. 4g, h, exhibiting a high electron mobility of up to 267 cm$^2$ V$^{-1}$ s$^{-1}$. The off-current is lower than 0.01 nA and the on-state current reaches 645 μA at $V_{DS}$ of 2 V and $V_G$ of 3 V, resulting in a large on/off current ratio ($I_{ON}/I_{OFF}$) of >10$^6$. The high on-state current is attributed to the multiple high-quality Bi$_2$O$_2$Se fins, indicating that uniform multiple-fin arrays can work independently and collaboratively. It is worth noting that the native-oxide Bi$_2$SeO$_5$ can act as the sole gate dielectric and enable remarkable gate control for the long channel 2D FinFETs (Supplementary Fig. 11). To obtain high-performance 2D FinFETs by shrinking channel length, HfO$_2$ layer was introduced into the devices. The additional HfO$_2$ layer acted as an insulating spacer to isolate the source/drain and gate electrodes and also served as dielectric because of its high dielectric constant ($\kappa \approx 16$).

For different transistors, the electrical properties are similar, indicating the great reliability and reproducibility of 2D multi-fin FETs (Supplementary Fig. 13). The average on-state current ($I_{on}$) is 760 ± 60 uA, mobility ($\mu$) is 165 ± 20 cm$^2$ V$^{-1}$ s$^{-1}$, and subthreshold swing ($SS$) is 200 ± 40 mV dec$^{-1}$, respectively. Note that the as-fabricated 2D FinFETs exhibit superior durability with almost no degradation in its electrical performance after performing repeated measurements for 50 times (Fig. 4h). The on/off current ratio remains almost constant and the threshold voltage ($V_{th}$) remains stable at around −0.46 V with a slight shift about 0.02 V in transfer curves (Fig. 4h, i). Remarkably, under the operating temperature of around 400 K, the multi-fin FinFET can still maintain an on/off ratio of more than 10$^5$, and the slightly increased off-state current can be recovered after cooling down to room temperature (Fig. 4j, k).

In order to evaluate the performance and energy efficiency of the 2D multi-fin FETs against that of the traditional Si and newly developed 2D semiconductor counterparts, we compared the normalized current of the fabricated 2D multi-fin FETs with 2D MoS$_2$ FinFETs and Intel's 14 nm-node Si FinFETs. Notably, owing to the different feature sizes of those transistors, the normalized current per level is employed as the comparison parameter to remove the effect of size. As illustrated in Fig. 4l, the drive current increases with the increase of $V_{DS}$. The 2D multi-fin FET can achieve a larger drive current per level of up to 230 μA μm μm$^{-1}$ ($V_{DS}$ = 1 V) under relative low gate-voltage modulation, surpassing that of 2D MoS$_2$ FET[49], 2D InSe FET[50], 2D MoS$_2$ FinFET[20] and Si-based FinFET[51], and revealing a potential for high-performance applications. More noteworthy is that 2D-semiconductor-based FinFET shows a strong advantage at energy efficiency that is remarkable for electronic devices with ultralow power consumption. The off-state current of as-fabricated 2D multi-fin FET is as low as 21.7 pA μm μm$^{-1}$ ($V_{DS}$ = 1 V), which is comparable with 2D MoS$_2$ FinFET and only 9.4% of that of Si-based FinFET.

To investigate the effect of fin number on 2D FinFET performance, we constructed 1-fin and 2-fin FETs using adjacent fins (Fig. 5a). In comparison with the 1-fin FET, the 2-fin FET exhibits a significant increase in on-state current, while maintaining superior electrostatic gate control (Fig. 5b and Supplementary Fig. 14). Meanwhile, the transconductance of the 2-fin FET is 1.7 times that of the 1-fin FET

(Fig. 5c). As the number of fins increases further to 5, the FinFETs demonstrate even larger on-state current and transconductance, indicating improved drive capability (Fig. 5d and Supplementary Fig. 14). In particular, the multi-fin FET with 5 fins delivers a high on-state current of up to 1 mA (Fig. 5e). The on-state current not only represents the drive capability, but also determines the intrinsic gate delay of transistors, which means the switching speed. Such high on-state current in 2D multi-fin FETs provides low gate delay of about 30 ps (Fig. 5f and Supplementary Table 1), which is comparable to Si-based metal-oxide-semiconductor (MOS) FETs at similar channel length. These remarkable performances of 2D muti-fin FETs highlight the potential of integrated 2D Bi$_2$O$_2$Se/Bi$_2$SeO$_5$ fin-oxide heterostructures as a promising candidate for next-generation advanced technology nodes.

## Discussion

In summary, we developed ledge-guided epitaxy as a versatile approach for the preparation of high-density mono-oriented 2D Bi$_2$O$_2$Se fin arrays on various insulating substrates. We demonstrated that atomically sharp steps of growth substrates play a crucial role in controlling the nucleation sites and in-plane orientation of vertical 2D Bi$_2$O$_2$Se fins. As-fabricated 2D multi-fin FETs based on the epitaxially integrated 2D Bi$_2$O$_2$Se/Bi$_2$SeO$_5$ fin-oxide arrays exhibit high on-state current and remarkable device durability, even during repeated measurements and at high operating temperatures. By further optimizing the preparation of high-density aligned steps with precise spacing control by ion beam etching, it is possible to achieve ordered higher-density 2D Bi$_2$O$_2$Se fin arrays. This advancement will facilitate large-scale integration of 2D multi-fin FETs, thus allowing for further 2D transistor scaling.

## Methods

### Preparation of substrates with high-density steps

The growth substrates of LaAlO$_3$ (100), MgO (110), and CaF$_2$ (110) single crystals are scratched by diamond scraper to fabricate high-density aligned steps. The step direction is determined by the crystal structure of the substrate.

### Synthesis of high-density aligned 2D Bi$_2$O$_2$Se fin arrays

High-density aligned 2D Bi$_2$O$_2$Se fin arrays were synthesized in a homemade chemical vapor deposition (CVD) system. Bi$_2$O$_3$ powder (Alfa Aesar, 99.999%) was placed in hot center of the furnace and Bi$_2$Se$_3$ powder (Alfa Aesar, 99.999%) was placed upstream 3.5 cm. The heating temperature was controlled at 610−650 °C. The total pressure of the growth chamber was kept at 400 Torr with the high-purity Ar gas serving as carrier gas, whose flow rate was 100 sccm (sccm: standard-state cubic centimeter per minute). The growth substrates were directly placed above the intersection of Bi$_2$O$_3$ and Bi$_2$Se$_3$ precursors with a gap of ∼4 mm. The growth time was 10−60 s.

### Characterizations

The morphology of as-synthesized aligned 2D Bi$_2$O$_2$Se fin arrays was characterized by scanning electron microscopy (SEM, Hitachi S4800 field emission electron microscope). The tilted SEM images were

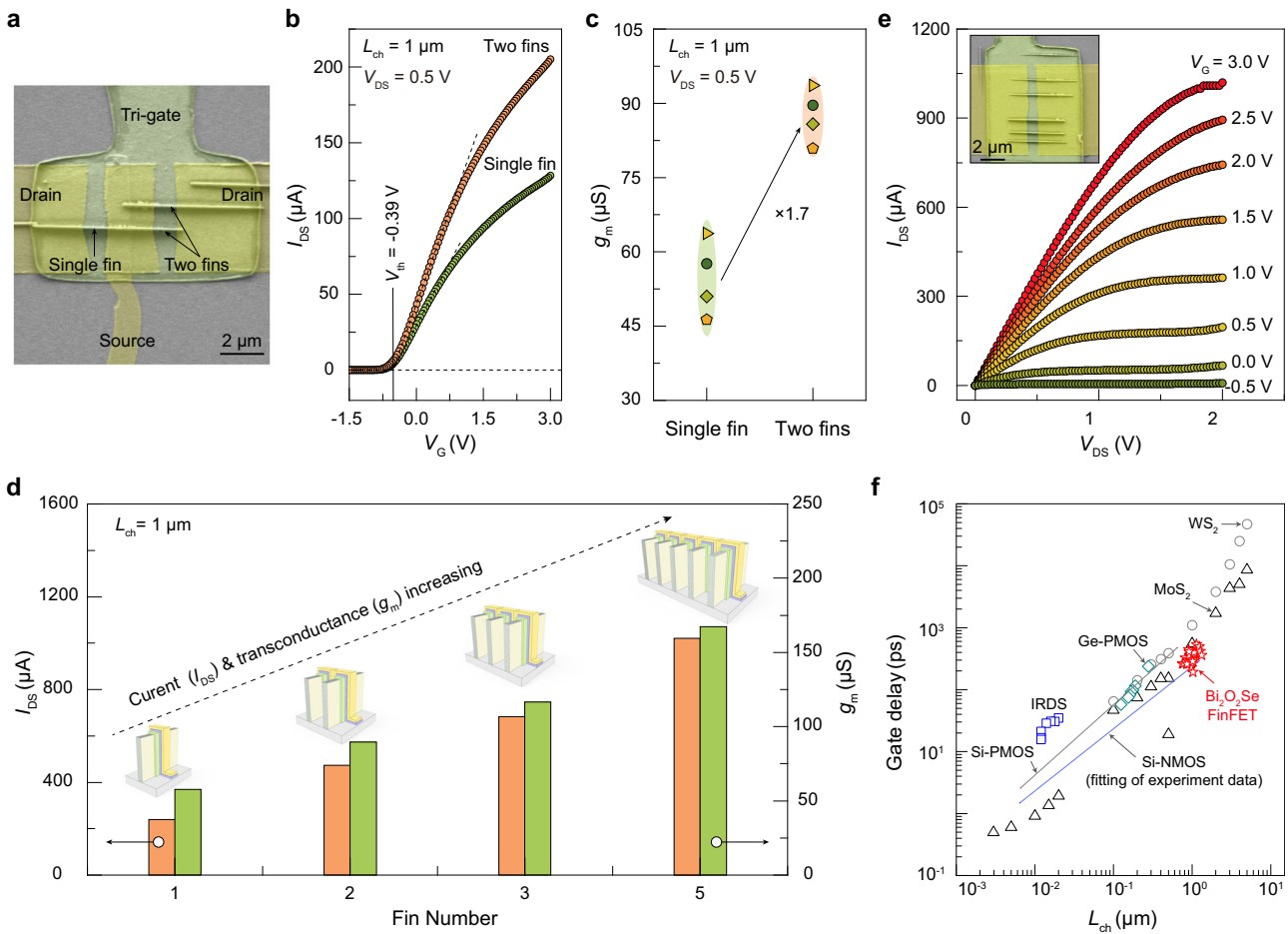

**Fig. 5 | Comparison of the electrical performances of 2D FinFETs with different number of fins. a** Top SEM image of FinFETs with single fin and two fins, respectively, which share one fin. **b** Transfer curves of 2D FinFETs with single fin and two fins. **c** Comparison of the transconductance of 2D FinFETs with single fin and two fins. The different data symbols were obtained from different devices. **d** Comparison of on-state current and transconductance of 2D FinFETs with different number of fins, demonstrating that multiple-channel FinFETs possess higher electrical performance. **e** Typical output curves of the 2D FinFETs with five fins. **f** Benchmarking of the gate delay of 2D multi-fin FETs versus the channel length ($L_{ch}$) with Si MOS[57], Ge MOS[58], $MoS_2$ FET[59–61] and $WS_2$ FET[59] (Part of data of $MoS_2$ and $WS_2$ are calculated from ref. 59,60). IRDS 2017–2033 targets[62] for high-performance (HP) transistors are also plotted.

obtained by means of a sample holder tilted at 45° angle. To characterize the relationship between the location of 2D fins and the steps, the guided 2D $Bi_2O_2Se$ fins were characterized using the atomic force microscopy (AFM, Bruker dimension icon, ScanAsyst mode).

## Cross-section STEM characterization of fin-substrate interface and FinFETs

The fin was thinned by focused ion beam (FEI Scios 2 Dual Beam SEM/FIB system) and cross-section TEM sample was obtained for characterizing the interface structure. Then the atomical structure of $Bi_2O_2Se$ fin/$LaAlO_3$ interface can be clearly demonstrated by an aberration-corrected scanning transmission electron microscope (AC-STEM) (FEI Titan Cubed Themis G2 300, operated at 300 kV acceleration voltage). The cross-section structure of FinFETs was also characterized by the same method.

## Density functional theory (DFT) calculations

All calculations were carried out using density functional theory (DFT) via the Vienna ab initio simulation Package (VASP)[52,53]. The projector augmented wave method was employed to describe the interaction between valence electrons and nuclei[54], and the Perdew–Burke–Ernzerhof (PBE) functional with generalized gradient approximation (GGA) was utilized to describe the exchange-correlation interaction[55]. The DFT-D3 dispersion-correction method[56]

was adopted to describe the van der Waals interactions. A kinetic energy cutoff of 500 eV was used for the plane wave basis set. Energy convergence criteria for electronic and ionic iterations were set to be $10^{-5}$ and $10^{-4}$ eV, respectively. A vacuum layer was set as ~15 Å.

The binding energy between a Bi/Se atom/Bi-O monomer/2D $Bi_2O_2Se$ fin nucleus and the $LaAlO_3$ substrate at different nucleation sites is defined as:

$$E_b = E_t - E_{sub} - E_{Bi/Se/BiO/Bi_2O_2Se} \tag{1}$$

Where $E_t$, $E_{sub}$ and $E_{Bi/Se/BiO/Bi_2O_2Se}$ are the total energy, the energy of the substrate, and the energy of Bi/Se/Bi-O/$Bi_2O_2Se$ adsorbed on the substrate, respectively.

The relative stabilities of $Bi_2O_2Se$, $Bi_2SeO_5$, and $O_2$ are compared by their formation energies, which are calculated by using

$$E_f = \frac{E_t}{N} \tag{2}$$

where $E_t$ is the total energy, $N$ is the number of atoms.

## Fabrication and measurements of 2D FinFETs

Aligned 2D fin arrays were used to fabricate FinFETs via the following device fabrication process. Firstly, in order to prevent 2D fin arrays

from collapsing during processing, patterned Au film was adopted to encapsulate 2D $Bi_2O_2Se$ fins. The pattern was processed with electron beam lithography (EBL) and the metal deposition (Au, 100 nm) was performed by magnetron sputtering coater (QAM-4W-STS, ULVAC) subsequently. Then, the channel windows were exposed by EBL processes and wet chemical etching to remove Au film. The etchant was an aqueous solution consisting $I_2$, KI, and $H_2O$ in the ratio of 0.5:1:30. To remove the etchant residues, the slices were quickly transferred into hot water (90 ℃) after being etched for 4–8 s. When the Au film on the channel window was completely etched, the remaining Au film acted as source and drain electrodes.

Next, high-$\kappa$ native-oxide gate $Bi_2SeO_5$ (5.5 nm) prepared by intercalative oxidation and high-$\kappa$ dielectric $HfO_2$ (7.2 nm) deposited by atomic layer deposition (ALD) were used as the dielectrics. Finally, the top-gated electrodes were achieved by EBL exposure and subsequent deposition of Ti/Au films (50 nm/100 nm).

The electrical measurements of the as-fabricated 2D $Bi_2O_2Se$ FinFETs were carried out by a Keithley SCS-4200 semiconductor parameter analyzer combined with a micromanipulator 6200 probe under ambient conditions.

### Strain mapping
Strain mapping was estimated according to the displacement of bright spots in the STEM image shown in Fig. 2. The strain ($\varepsilon_{xx}$) was calculated by a peak-pair algorithm, and the formula is as follows:

$$\varepsilon_{xx} = \frac{\partial u}{\partial x} = \frac{\partial\left(\sqrt{u_{xx}^2 + u_{yy}^2}\right)}{\partial x} \tag{3}$$

Where $u_{xx}$ and $u_{yy}$ are the displacements of the bright spots in the in-plane <001> direction and the vertical <100> direction, respectively. $u_{xx}$ and $u_{yy}$ are calculated by $u_{xx} = \Delta x - a_{LaAlO_3(x)}$ and $u_{yy} = \Delta y - a_{LaAlO_3(y)}$. Here $\Delta x$ and $\Delta y$ represent the displacements of the bright spots in each direction, and $a_{LaAlO_3(x)}$ and $a_{LaAlO_3(y)}$ are lattice constants of the in-plane and vertical direction of the $LaAlO_3$ (100) substrate.

### Calculations of field-effect mobility and intrinsic gate delay for 2D $Bi_2O_2Se$ FinFETs
The field-effect mobility of $Bi_2O_2Se$ FinFETs was extracted from the linear region of transfer curves. The field-effect electron mobility was calculated according to the following relations:

$$\mu_{FET} = \frac{W_{eff}}{L_{ch}} \times \frac{1}{C_G^*} \times \frac{\partial I_{DS}}{V_{DS}\partial V_G} \tag{4}$$

$$C_G^* = \frac{C_G(HfO_2) \times C_G(Bi_2SeO_5)}{C_G(HfO_2) + C_G(Bi_2SeO_5)} \tag{5}$$

$$C_G = \frac{\varepsilon_r\varepsilon_0}{d} \tag{6}$$

Where $C_G$ is the top-gate oxide capacitance, $L_{ch}$ is the channel length of the devices, $W_{eff}$ represents the effective width of the devices, i.e., the sum of the width and twice the height of all 2D $Bi_2O_2Se$ fins ($W_{eff}$ = (fin width + 2 × fin height) × fin number). The $\varepsilon_r$ for $Bi_2SeO_5$ and $HfO_2$ is adopted 21 and 16, respectively. Here, the tri-gate dielectrics are composed of 5.5 nm-thick $Bi_2SeO_5$ and 7.2 nm-thick $HfO_2$, the $C_G^*$ equals $1.243 \times 10^{-2}$ F m$^{-2}$.

The intrinsic gate delay is defined as[57]

$$\tau = \frac{CV_{DD}}{I_{on}} \tag{7}$$

$$C = C_G^* \times W_{eff} \times L_{ch} \tag{8}$$

in which $V_{DD}$ is the supplied voltage of operation and $I_{on}$ is the on-state current. $C$ is the total gate capacitance.

## Data availability
Relevant data supporting the key findings of this study are available within the article and the Supplementary Information file. All raw data generated during the current study are available from the corresponding authors upon request.

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

## Acknowledgements

We acknowledge the Molecular Materials and Nanofabrication Laboratory (MMNL) in the College of Chemistry at Peking University for the use of instruments. This work was supported by the National Natural Science Foundation of China (21920102004, 22205011, and 92164205), the National Key Research & Development Program of China (2021YFA1202901), Beijing National Laboratory for Molecular Sciences (BNLMS-CXTD-202001), and the New Cornerstone Science Foundation through the XPLORER PRIZE.

## Author contributions

H.P., C.T. and M.Y. conceived for the project and designed the experiments. M.Y. and C.T. performed the synthesis of high-density 2D fin arrays guided by steps and corresponding SEM characterizations. M.Y.,

C.T. and X.G. conducted the STEM and analyzed the results. C.T. and J.T. performed the device fabrication and electrical characterization. F.D. and Y.Y. conducted the theoretical calculations. H.P. supervised this research. M.Y., C.T. and H.P. co-wrote the manuscript. H.L. and other authors contributed to discussions.

## Competing interests

The authors declare no competing interests.
