## [Peer Review File · Nature Communications]

Integrated 2D multi-fin field-effect transistorsREVIEWER COMMENTS

Reviewer #1 (Remarks to the Author):

The manuscript entitled “Vertical fin of semiconducting channel integrated with high- κ oxide dielectric in advanced transistor scaling” by Yu et al. provides insights into the development and fabrication of advanced transistor structures, specifically focusing on the utilization of 2D semiconductors in a vertical fin architecture. Integrating single fin channels based on two-dimensional (2D) semiconductors offers the potential of achieving sub 1 nm fin-width, a feat that has eluded current Si-based FinFETs. This promises superior performance and higher integration density. Specifically, using a ledge-guided epitaxy strategy is a novel approach, offering a pathway to grow high-density, mono-oriented 2D Bi₂O₂Se fin arrays. In light of the practical utility, the developed method showcases how to enhance the binding of 2D fin nuclei at the step edge, which has implications for precise nucleation and orientation of fin arrays. The fabricated multi-channel 2D FinFETs have shown reasonable electrical performances, especially regarding on/off current ratios and durability at high temperatures. While I am leaning toward publication in Nature Communications, the authors should delve deeper into the growth mechanism, specifically detailing the ledge-guided epitaxy strategy's nuances. This would provide clarity on its novelty and utility in producing high-density 2D fin arrays, distinguishing the current results from their prior work (Nature 616, 66 (2023)).

1. The Lack of Growth Mechanism(s). It is unclear how the authors produced artificially manufactured ledge-guided epitaxy strategy on oxide substrates. The overly simplified description falls short of elucidating the actual growth mechanisms. A more in-depth explanation is necessary to understand how this process differs from existing methods and its unique advantages. Meanwhile, some important references were omitted in the original submission, including Nat. Nanotechnol. (2023).
<https://doi.org/10.1038/s41565-023-01445-9> and Matter (2023)
<https://doi.org/10.1016/j.matt.2023.05.034>
2. Wafer-scale Selective Growth. OM and SEM images that demonstrate the periodic arrays of 2D fin-oxide with control over spacing and thickness of fins shall be included.
3. Median Output Performance: While the authors emphasize the superior performance of the transistors, the median output performance is not convincing. This is crucial for comprehensively evaluating the transistors' reliability and reproducibility. Further, short channel devices made of multi-Fin channels (>3) shall be included. Transfer characteristics and statistical analyses of subthreshold swing (SS), on-current, and mobility should be provided.
4. The author should provide a clearer comparison between the traditional Si-based FinFETs and the newly developed 2D semiconductor-based FinFETs in terms of performance, energy efficiency, and integration density.
5. There are many typos in the body of the manuscript and figure caption. For example, in Figure 5, a comparison of the “enectrical” performances. The authors should carefully go through the manuscript during the revision.

Reviewer #2 (Remarks to the Author):

In this manuscript, a novel synthesis and device fabrication approach was developed to produce finFET based off a new kind of 2D semiconductors, Bi₂O₂Se. 2D Bi₂O₂Se single crystal flakes were grown vertically on various substrates, including LaAlO₃, MgO, and CaF₂. The authors engineered the substrate surface and take advantage of the terrains on the miscut surfaces of these substrate to guide the growth of the 2D semiconductor fins. The authors further demonstrated finFET devices made with the vertical B₂O₂Se flakes and carried on the electrical measurements of the devices. Although 2D semiconductors have shown great promise in further facilitating the scaling of transistors, most of the previous studies on this topic have been focused on planer 2D semiconductor transistor architecture. Only recently have there been very few investigations on 3D transistor architectures (including finFETs, nanosheet FETs, and CFETs). The synthesis approach for crystalline vertical 2D semiconductors is innovative, and could be an encouraging technology enabler for various research and development activities for 3D transistor architectures using the emerging family of 2D semiconductors. The dimensions of the finFET devices shown in this manuscript are still too large to show the benefits of electrostatic controllability and crystallinity of using 2D semiconductors as an ultimately scaled transistor solution. It would be nicer to see the finFET made with 2D semiconductors with less than 5 nm fin thickness, and benchmark their device performance with the silicon counterparts. Nevertheless, this work is a necessary milestone toward the right direction of the technology development for 2D semiconductor transistor technologies, and would be of general interest to the readership of Nature Communications.

Below are my detailed technical questions/comments:

- (1) I am curious to see the feasibility of growing Bi₂Se₂O vertical fins with thickness around or smaller than 5 nm. Then it is going to be a fair comparison to silicon finFETs. If it is challenging, I would like to learn from the authors what the challenges are and if there are any viable pathway to overcome the challenges.
- (2) The authors deposited additional layers of HfO₂ dielectrics, in addition to the native oxide Bi₂Se₂O layers, as the gate stack of their finFET technology. Please provide the reasons why the additional HfO₂ layer is necessary, and if possible, please shown device data without the HfO₂ layer.
- (3) The crystal structures of the native oxide layer on the top edge of the B₂Se₂O fins look very different from that on the side surfaces. Please include some description and characterization to compare the two regions and comment on how would such discrepancy may impact the finFET I-V characteristics.
- (4) The authors provided a novel approach to engineer the surface terrains of the growth substrate, which can guide the growth orientation of the Bi₂Se₂O vertical flakes. However, there are still variations in terms of the fin thickness, fin spacing, etc. I was wondering if there is a way to better control the placement and size distributions of the Bi₂Se₂O crystals.
- (5) In the introductory paragraphs, the authors should cite early works for the development of silicon finFET technologies. The authors should also include pervious studies on 3D transistor architectures made with 2D semiconductors (finFETs, nanosheet FETs, and CFETs).

Reviewer #3 (Remarks to the Author):

The authors presented their work on fabricating (multiple) fins primarily made on Bi₂O₂Se/Bi₂SeO₅ fin-oxide heterostructures, with the fins being grown vertically, through ledge-enhanced epitaxy on LaAlO₃-terminated surfaces. The authors devote a part of their work to explaining the fabrication process, then explaining the enhancement of the nucleation at the ledge site with support of DFT calculations and, finally, demonstration of finfet devices made with such fin-oxide heterostructures, eventually in a multifin configuration. The devices show good performance, at least in terms of current drive and, taken together, this is a reasonable work, despite several technical inaccuracies, which I will elaborate on later in this review.

Despite all these considerations, I believe the work is not suitable for being published Nature Communications, and I will present below my arguments:

1) When we put this work in context, faced with existing literature, it appears to me that the paper referenced as [15], Tan C. et al, and recently published in Nature (2023), by the same group (first two authors in reversed order), already disclosed most of the findings reported here. Namely: fabrication of the fin, possibility to control their (unidirectional) formation through techniques similar to what is discussed here, and even reported drive currents and on/off current ratio superior to those reported here: i.e. 830uA/um, on/off of 1e7, etc. When all these are taken into account, the added value of this work is practically reduced to the understanding with the help of DFT. This is, in my opinion, too thin to defend eligibility for Nat. Comm. publication. To be clear: even if there are technical inaccuracies in this work (I will explain), the fundamental issue I have with the publication of this work is the overall novelty and added value for a journal of the caliber of Nat. Comm. I would see no problem in considering this work and improving it further in view of publication in other journals where the emphasis is, for instance, on providing a more in depth understanding and systematic analysis of an already reported novel concept.

2) Technical inaccuracies and weaknesses of this work: regardless of my previous point, I have several remarks regarding this work, and outline below a few:

- the fins are thick: 18-24 nm in thickness, that means a multilayer structure. 2D materials are appealing because they are shown to perform at monolayer thickness, a regime that is clearly inaccessible to Si, where severe degradation of the electrical performance onsets at around 5-6nm channel thickness. I think the challenges of this guided (vertical) growth would be huge when trying to approach control of the fin width. Could the authors grow these fins a few monolayers (not to say 1 monolayer) thin? I think this is hitting in fact another key problem connecting to the mechanical stability, which would actually lead to considering fins laid horizontally, not standing vertically (these first ones are also called "nanosheets" in the technologists' slang)
- the authors use repeatedly the term "integration". I do not see much integration here; there is a long, long way from these structures to integrated devices in the common acceptance of the term, used in the semiconductor tech industry and R&D.
- The authors report that their fin arrays could "meet the requirement of advanced 3-nm technology node, as projected by the International Roadmap for Devices and Systems" (p.5, 190-192). As a matter of

fact, the 3-nm node is in production this year, and not by one manufacturer, but 2, and another one on its way. Then, why would it be needed to do research with (still) exotic materials for something that's available commercially? If IRDS is brought into the picture, I would expect to focus on the nodes that are still in R&D, and have not yet solutions available in all areas, and where 2D materials are expected to make a difference. These would be 1nm eq. and beyond, with a time horizon for reaching industrial maturity in the next decade.

These being said, I regret for not being able to recommend this work for publication in Nat.Comm.

REVIEWER COMMENTS

Reviewer #1 (Remarks to the Author):

The manuscript entitled “Vertical fin of semiconducting channel integrated with high- κ oxide dielectric in advanced transistor scaling” by Yu et al. provides insights into the development and fabrication of advanced transistor structures, specifically focusing on the utilization of 2D semiconductors in a vertical fin architecture. Integrating single fin channels based on two-dimensional (2D) semiconductors offers the potential of achieving sub 1 nm fin-width, a feat that has eluded current Si-based FinFETs. This promises superior performance and higher integration density. Specifically, using a ledge-guided epitaxy strategy is a novel approach, offering a pathway to grow high-density, mono-oriented 2D Bi₂O₂Se fin arrays. In light of the practical utility, the developed method showcases how to enhance the binding of 2D fin nuclei at the step edge, which has implications for precise nucleation and orientation of fin arrays. The fabricated multi-channel 2D FinFETs have shown reasonable electrical performances, especially regarding on/off current ratios and durability at high temperatures. While I am leaning toward publication in Nature Communications, the authors should delve deeper into the growth mechanism, specifically detailing the ledge-guided epitaxy strategy's nuances. This would provide clarity on its novelty and utility in producing high-density 2D fin arrays, distinguishing the current results from their prior work (Nature 616, 66 (2023)).

Authors' response

We deeply appreciate the positive and insightful comments from the referee on the innovation of our work. The referee's constructive suggestions have helped us to improve the quality of our manuscript.

To unveil the ledge-guided epitaxy mechanism, we employed the first-principles density functional theory (DFT) in conjunction with experiments to further illustrate the preferred nucleation of 2D fin at the ledge and the guided growth along the step of substrate. The preferred alignment of 2D fin seeds suggests that the artificial ledges may represent preferential nucleation sites with minimal local energy, avoiding random nucleation in the absence of ledges (Nature 2023, 616, 66). We will fully address the reviewer's comments point by point in the following.

1) The Lack of Growth Mechanism(s). It is unclear how the authors produced artificially manufactured ledge-guided epitaxy strategy on oxide substrates. The overly simplified description falls short of elucidating the actual growth mechanisms. A more in-depth explanation is necessary to understand how this process differs from existing methods and its unique advantages. Meanwhile, some important references were omitted in the original submission, including Nat. Nanotechnol. (2023). <https://doi.org/10.1038/s41565-023-01445-9> and Matter (2023)

Authors' response

We are very thankful for the constructive comments, which helped us to make our work clearer.

In our work, the aligned 2D fins are grown epitaxially by employing a ledge-guided epitaxy strategy that relies on thermodynamic control of the seeding orientation of the 2D fins through ledge-induced selective nucleation. As shown in Figure R1, the ledge-guided epitaxy mainly involves the following four processes (taking LaAlO_3 as a representative example): (i) a single-crystal epitaxy substrate with exposed ledges is adopted; (ii) exposed ledges on the substrate surface preferentially trap precursor atoms and thereby serve as nucleation sites; (iii) 2D fin seeds with energetic minimum nucleate at the ledge, breaking the symmetry and selectively stabilizing a preferred orientation; (iv) mono-oriented seeds grow anisotropically into well-aligned 2D fins.

Figure Response 1. Schematic of ledge-guided epitaxy process, including steps formation, ledge-guided nucleation and growth.

Driven by the ledge-guided epitaxial mechanism, we can adopt two approaches to create aligned steps on the surface of epitaxial substrates. As shown in Figure R2, the first methodology involves artificially created parallel scratches on the substrate surface using a diamond scraper (Figure R2a). Owing to the different hardness values of the diamond and substrate (e.g. LaAlO_3 , MgO , CaF_2 etc.), the “scratched lines” with atomic resolution steps can be successfully generated with simple scratches. In addition, each scratched line consists of multiple parallel steps whose orientation is determined by the crystal lattice of the substrate. On LaAlO_3 (100) surface, all artificial steps are completely parallel to $[010]_{\text{LaAlO}_3}$ or $[001]_{\text{LaAlO}_3}$. These macroscopic steps have a height of approximately 10 nm and the step spacing range from tens of nanometers to a few hundred nanometers (Figure R2b, c). The microscopic characterization indicates that the macroscopic steps are composed of numerous atomical steps (Fig. 2b). Remarkably, after combining a micromachined arm and a diamond scrape, the spacing of 2D fin arrays is controllable by controlling the spacing of step arrays (Figure R2d, e).

Figure Response 2. Creating process of artificially parallel steps using a diamond scraper. **a**, Schematic illustration of creating line-array scratches by diamond scraper. **b, c**, AFM image (**b**) and corresponding step height (**c**) of the artificial steps. **d**, SEM image of guided-grown 2D fin arrays while using a micromachined arm and a diamond scrape to control the step spacing. **e**, Statistical orientations of the 2D fin arrays in (**d**).

Another strategy is to miscut the substrate with an miscut angle. Generally, as the miscut angle increases, the step density on the substrate surface increases simultaneously. The aligned steps throughout the substrate surface effectively tune the symmetry of the epitaxial surface. The epitaxial effect of miscut substrate is exemplified on miscut LaAlO_3 (100), where the symmetry of the LaAlO_3 (100) facet lowers from C_{4v} to C_{2v} as the miscut angle reaches $\sim 10^\circ$, yielding purely mono-oriented 2D $\text{Bi}_2\text{O}_2\text{Se}$ fin arrays (Figure R3).

Figure Response 3. Formation of parallel steps using miscut substrate. **a**, Schematic of miscutting the substrate to obtain aligned steps and control the orientation. **b**, SEM image of mono-oriented 2D fins on a miscut substrate.

Remarkably, the high-density steps from miscut angle can only tune the orientation of 2D fins with random nucleation, but they are unable to precisely control the nucleation

sites of 2D fins. With regards to the artificial steps from diamond scrape, they can control both the nucleation sites and orientation of 2D fins. Since these artificial steps contain numerous defects, firstly, they guide the nucleation of 2D fins at the step edges (Figure R4a). Then, the vertical 2D nuclei grow across the steps to form 2D fins. Density functional theory (DFT) calculations also reveal that lies on the superiority of the nucleation of the 2D Bi₂O₂Se fins at the step edges over that on terrace (Figure R4b-d). Exposed ledges on the substrate surface preferentially trap precursor Bi/Se atoms and Bi-O monomers and thereby serves as nucleation sites, lowering the binding energy of 2D Bi₂O₂Se fin nucleus at the step edge than that on the terrace. Furthermore, these artificial steps also lower the symmetry of substrates, anchoring the generated 2D fin seeds to grow along the parallel steps with mono-orientation.

Figure Response 4. Growth mechanism of ledge-guided epitaxy of 2D fins. **a**, Schematic of the nucleation and growth process of 2D fin at the step edge. **b-d**, The binding energies and optimized structures of Bi/Se atoms (b), Bi-O monomers (c) and a 2D Bi₂O₂Se nucleus (d) at the step edge and terrace, demonstrating that nucleation at the step edge is energetically favorable.

In our revised version, Figure R1 and R2 have been included as supplementary Fig. 4 and supplementary Fig. 2, respectively. Accordingly, we have revised “*The whole ledge-guided epitaxy mainly involves the following four processes (taking LaAlO₃ as a representative example): (i) a single-crystal epitaxy substrate with exposed ledges is adopted...*” in the main text on page 3-4. Additionally, the references “*Li, T., et al. Epitaxial growth of wafer-scale molybdenum disulfide semiconductor single crystals on sapphire. Nat. Nanotechnol. 2021, 16, 1201-1207. and Fu, J.-H., et al. Pieces of 2D materials: The next step to crystallize the polycrystalline domains. Matter 2023, 6, 2136-2152.*” have been added in revision text.

2) *Wafer-scale Selective Growth. OM and SEM images that demonstrate the periodic arrays of 2D fin-oxide with control over spacing and thickness of fins shall be included.*

Authors' response

We are very thankful for the constructive suggestions by the referee. Based on our understanding of the mechanism of ledge-guided epitaxy, we manage to fabricate periodic arrays of 2D fin-oxide with the assistance of artificial step arrays. There are indeed variations in terms of the fin thickness and fin spacing. The fluctuation of fin spacing primarily comes from the variation of step spacing. We admit that it is difficult to create a highly ordered step arrays via a diamond scrape. As shown in Figure R5, after combining a micromachined arm and a diamond scrape, the spacing of 2D fin arrays is controllable by controlling the spacing of step arrays. However, the thickness and spacing of as-grown 2D fins in single arrays is difficult to control. More efforts must be made to fabricate 2D fin arrays with more uniform morphology and uniform spacing in the future. For example, nanoimprinting can be employed to fabricate the periodic step arrays, which in turn enables control of the fin spacing finally.

Figure Response 5. 2D fin arrays with controllable fin spacing. a, SEM image of guided-grown 2D fin arrays while using a micromachined arm and a diamond scrape to control the step spacing. b, Statistical orientations of the 2D fin arrays in (a). Insert: schematic for the fin orientation.

Besides, the fluctuations in fin thickness are the result of kinetic factors, which can be resolved by constituent growth. As shown in Figure R6, we tried to combine maskless laser lithography system and Ar-based ion-beam etching technology to create a periodic nucleation-side arrays. The fin spacing is mainly determined by the step spacing, so as the step spacing is fixed, the fin spacing will fluctuate less. For large-area growth, the fin spacing is relatively uniform but the fin thickness fluctuates obviously (Figure R6e, f). After adjusting the growth conditions (such as the weight and ratio of precursors), we managed to obtain thinner 2D fin arrays with uniform thickness of 25 nm, but the fluctuation of fin spacing is relatively higher (Figure R7).

Figure Response 6. Periodic arrays of 2D fin-oxide heterostructures. **a**, Schematic illustration of the fabrication process of 2D fin-oxide periodic arrays. The pre-created step arrays contain numerous defects, which ensure lower nucleation energy barrier and thus anchor the 2D nuclei for the epitaxy of 2D periodic fin arrays. **b**, AFM image of artificially prepatterned step arrays on substrates. **c**, **d**, Typical SEM images of periodic arrangements of 2D fin (**c**) and 2D fin-oxide heterostructure (**d**) arrays. **e**, **f**, Low-magnification and high magnification SEM images of periodic arrays of 2D fin-oxide heterostructure.

Figure Response 7. Controllability of the fin spacing and thickness of 2D fin-oxide heterostructure arrays. **a**, Top view SEM image of periodic 2D fin-oxide heterostructure arrays with relatively uniform fin spacing and thickness. **b**, **c**, The fin spacing and

thickness fluctuation of 2D fin-oxide heterostructure arrays in (a).

Combining the above two methodologies, we are fully confident that a more controllable preparation of 2D fin arrays can be realized by using nano-imprinting technique. However, more efforts and time should be required, and we will struggle to achieve the preparation of 2D fin arrays with uniform spacing and thickness in future works based on this method.

3) Median Output Performance: While the authors emphasize the superior performance of the transistors, the median output performance is not convincing. This is crucial for comprehensively evaluating the transistors' reliability and reproducibility. Further, short channel devices made of multi-Fin channels (>3) shall be included. Transfer characteristics and statistical analyses of subthreshold swing (SS), on-current, and mobility should be provided.

Authors' response

We are very grateful for the constructive comments by the referee.

To characterize the reliability and reproducibility of 2D multi-fin FETs, we fabricated 10 2D 3-fin FETs on two different fabrication batches. As shown in Figure R8a, b, we show the transfer and output characteristics measured for 10 devices from two substrates, which shows similar electrical properties. The as-fabricated 2D multi-fin FETs showing off-state current (I_{off}) as low as 40 pA, on-off ratio ($I_{\text{on}}/I_{\text{off}}$) as high as 10^7 and on-state current (I_{on}) as high as 820 μA calculated from 10 devices in different fabrication batches. Figure R8c-e plot the statistical on-state current (I_{on}), mobility (μ) and subthreshold swing (SS) measured from these 10 devices with 1 μm channel length. The average I_{on} is $760 \pm 60 \mu\text{A}$, μ is $165 \pm 20 \text{ cm}^2 \text{ V}^{-1} \text{ s}^{-1}$, and SS is $200 \pm 40 \text{ mV dec}^{-1}$, respectively. Optimizing the device fabrication conditions to improve the subthreshold swing of the transistor is our next plan.

For short channel devices made of multi-fin channels, yet it is still challenging to apply the fabrication method into ultra-scaled devices. For example, the volatility of the wet etching involved in the device processing flow is still relatively high. We will continue working on this project to reach the goal of ultra-scaled 2D multi-fin FET with short channel length in the future work.

In our revised version, Figure R8 has been included as supplementary Fig. 13. Accordingly, we have revised “*For different transistors, the electrical properties are similar, indicating the great reliability and reproducibility of 2D multi-fin FETs (Supplementary Fig. 13). The average on-state current (I_{on}) is $760 \pm 60 \mu\text{A}$, mobility (μ) is $165 \pm 20 \text{ cm}^2 \text{ V}^{-1} \text{ s}^{-1}$, and subthreshold swing (SS) is $200 \pm 40 \text{ mV dec}^{-1}$, respectively.*”

in the main text, page 7.

Figure Response 8. Device-to-device variability. **a, b**, Transfer (a) and output (b) characteristics of 10 2D multi-fin FETs (the fin number is 3). **c-e**, Statistical distributions of on-state current (c), mobility (d), and SS (e) for 10 devices.

4) *The author should provide a clearer comparison between the traditional Si-based FinFETs and the newly developed 2D semiconductor-based FinFETs in terms of performance, energy efficiency, and integration density.*

Authors' response

We are very thankful for the constructive comments by the referee.

To evaluate the performance and energy efficiency of the 2D multi-fin FETs against that of the traditional Si and newly developed 2D semiconductor counterparts, we compared the normalized current of the fabricated 2D multi-fin FETs with 2D MoS₂ FinFETs (M. Chen *et al.*, TMD FinFET with 4 nm thin body and back gate control for future low power technology, 2015 *IEEE International Electron Devices Meeting (IEDM)*, 32.32.31-32.32.34) and Intel's 14nm-node Si FinFETs (S. Natarajan *et al.*, A 14nm Logic Technology Featuring 2nd-Generation FinFET, Air-Gapped Interconnects, Self-Aligned Double Patterning and a 0.0588 μm² SRAM cell size, 2014 *IEEE International Electron Devices Meeting (IEDM)*, 3.7.1-3.7.3). Notably, owing to the different feature sizes of those transistors, the normalized current per level is employed as the comparison parameter to remove the effect of size. As illustrated in Figure R9, the drive current increases with the increase of V_{DS} . The 2D multi-fin FET can achieve a larger drive current per level of up to 230 μA μm μm⁻¹ ($V_{DS} = 1$ V), surpassing that of Si-based FinFET and 2D MoS₂ FinFET, and revealing a potential for high-performance applications. More noteworthy is that 2D-semiconductor-based FinFET show a strong

advantage at energy efficiency, which is remarkable for electronic devices with ultralow power consumption. The off-state current of as-fabricated 2D multi-fin FET is as low as $21.7 \text{ pA } \mu\text{m } \mu\text{m}^{-1}$ ($V_{\text{DS}} = 1 \text{ V}$), which is comparable with 2D MoS₂ FinFET and only 9.4% of that of Si-based FinFET. In summary, the 2D multi-fin FETs exhibits promising in both high-performance and low-power consumption small footprint electronic applications.

In our revised version, Figure R9 has been included in Fig. 4. Accordingly, we have revised “*In order to evaluate the performance and energy efficiency of the 2D multi-fin FETs against that of the traditional Si and newly developed 2D semiconductor counterparts, we compared the normalized current of the fabricated 2D multi-fin FETs with 2D MoS₂ FinFETs and Intel’s 14nm-node Si FinFETs...*” in the main text, page 7.

Figure Response 9. Comparison of normalized current of the fabricated 2D multi-fin FETs with 2D MoS₂ FinFETs and Intel’s 14nm-node Si FinFETs.

5) *There are many typos in the body of the manuscript and figure caption. For example, in Figure 5, a comparison of the “enectrical” performances. The authors should carefully go through the manuscript during the revision.*

Authors’ response

We are very grateful to the reviewer for pointing out our spelling mistakes. We have corrected the mistake in the caption of Figure 5 in our revised version and carefully gone through the whole manuscript.

Reviewer #2 (Remarks to the Author):

In this manuscript, a novel synthesis and device fabrication approach was developed to produce finFET based off a new kind of 2D semiconductors, Bi₂O₂Se. 2D Bi₂O₂Se single crystal flakes were grown vertically on various substrates, including LaAlO₃, MgO, and CaF₂. The authors engineered the substrate surface and take advantage of the terrains on the miscut surfaces of these substrate to guide the growth of the 2D semiconductor fins. The authors further demonstrated finFET devices made with the vertical Bi₂O₂Se flakes and carried on the electrical measurements of the devices. Although 2D semiconductors have shown great promise in further facilitating the scaling of transistors, most of the previous studies on this topic have been focused on planer 2D semiconductor transistor architecture. Only recently have there been very few investigations on 3D transistor architectures (including finFETs, nanosheet FETs, and CFETs). The synthesis approach for crystalline vertical 2D semiconductors is innovative, and could be an encouraging technology enabler for various research and development activities for 3D transistor architectures using the emerging family of 2D semiconductors. The dimensions of the finFET devices shown in this manuscript are still too large to show the benefits of electrostatic controllability and crystallinity of using 2D semiconductors as an ultimately scaled transistor solution. It would be nicer to see the finFET made with 2D semiconductors with less than 5 nm fin thickness, and benchmark their device performance with the silicon counterparts. Nevertheless, this work is a necessary milestone toward the right direction of the technology development for 2D semiconductor transistor technologies, and would be of general interest to the readership of Nature Communications.

Authors' response

We deeply appreciate the referee's positive comments on the innovation and quality of our work. The constructive suggestions from the referee help bring significant improvements to our manuscript. We noticed that the key concerns are related to the fabrication of 2D FinFET based on fin with a thickness less than 5 nm and the device performance comparison with silicon. Combining controlled surface oxidation of 2D fins, 2D fin can be thinned to 3 nm, which is expected to fabricated high-performance 2D FinFET with ultra-thin fin. Our point-by-point responses are listed below.

Below are my detailed technical questions/comments:

1) I am curious to see the feasibility of growing Bi₂O₂Se vertical fins with thickness around or smaller than 5 nm. Then it is going to be a fair comparison to silicon finFETs. If it is challenging, I would like to learn from the authors what the challenges are and if there are any viable pathway to overcome the challenges.

Authors' response

We are very thankful for referee's constructive question. We agree with the referee that growing 2D Bi₂O₂Se fins with thickness smaller than 5 nm is critical for fair comparison to silicon FinFETs.

Generally, the growth rate of 2D Bi₂O₂Se fins is ultrafast. As shown in Figure R8, the thickness of 2D fins can be tuned by changing the oxygen concentration. With increasing oxygen, the thickness of as-grown 2D Bi₂O₂Se fins was increased (Figure R10). The reason for the above phenomenon is likely related to the oxygen absorption on substrate surface in nucleation process of 2D fins. When the oxygen concentration is relatively high, the absorbing rate of the oxygen is relatively high on the substrate surface. Thereby, the absorbed precursors would accumulate and nucleate on the substrate with more probability, resulting in crystallizing into relatively thick 2D Bi₂O₂Se fins. By controlling the low oxygen concentration and short growth time (~ 10 s), the thickness of 2D fins can reach around 18 nm. Because of the ultra-fast growth rate in current preparation strategy, it is challenging to directly grow 2D fins with thickness smaller than 5 nm.

Figure Response 10. Control of the thickness of 2D Bi₂O₂Se fins. **a-c**, SEM images of fins obtained under 0, 20 and 40 ppm O₂, respectively. The growth time is ~10s. **d**, Statistics for fin thickness as a function of oxygen concentration.

Nevertheless, intercalative oxidation is an efficient method to thinning 2D Bi₂O₂Se fins (Figure R11). As the thickness of oxidized Bi₂O₂Se can be well-controlled by time and temperature (Y. Zhang, H. Peng *et al.* A single-crystalline native dielectric for two-dimensional semiconductors with an equivalent oxide thickness below 0.5 nm, *Nature Electron.* 2022, 5, 643–649), ultra-thin 2D Bi₂O₂Se fins can be obtained after suitable oxidization process. The oxidization process does not destroy the pristine morphology of 2D fins (Figure R11b). As shown in Figure R11c, 2D Bi₂O₂Se fin can be thinned to 3 nm through controlled intercalative oxidation.

In our revised version, Figure R11 has been included in supplementary Fig. 8. Accordingly, we have revised “*On the one hand, controllable oxidation is an effective approach to thinning 2D fins in the fin-oxide heterostructures, which enables 2D fin as thin as 3 nm (Supplementary Fig. 8e)*” in the main text, page 6.

Figure Response 11. Fabrication of ultra-thin 2D $\text{Bi}_2\text{O}_2\text{Se}$ fin. **a.** Schematic illustration for preparing ultrathin fins by intercalative oxidation. **b.** Tilted-view SEM images of as-synthesized rectangular 2D $\text{Bi}_2\text{O}_2\text{Se}$ fin arrays (left) and $\text{Bi}_2\text{O}_2\text{Se}/\text{Bi}_2\text{SeO}_5$ fin-oxide heterostructure arrays after intercalative oxidation (right) on the MgO (110) surface. **c.** Cross-sectional HR-STEM images of the 2D $\text{Bi}_2\text{O}_2\text{Se}$ fin showing the ultrathin body of about 3 nm.

2) The authors deposited additional layers of HfO_2 dielectrics, in addition to the native oxide Bi_2SeO_5 layers, as the gate stack of their finFET technology. Please provide the reasons why the additional HfO_2 layer is necessary, and if possible, please shown device data without the HfO_2 layer.

Authors' response

We are grateful for the referee's concerns. Actually, the dielectric HfO_2 layer is not necessary for the long-channel 2D FinFET devices. As shown in Figure R12, we managed to fabricate the 2D FinFETs with a channel length of $3.5 \mu\text{m}$, in which the epitaxial native-oxide Bi_2SeO_5 was directly used as gate dielectric. Without using additional HfO_2 layer, the fabricated 2D $\text{Bi}_2\text{O}_2\text{Se}/\text{Bi}_2\text{SeO}_5$ FinFETs illustrates an excellent gate control with an on-off ratio of $>10^6$.

To fabricate high-performance 2D FinFETs by shrinking channel length, the HfO_2 layer is critical in our fabrication protocol. The HfO_2 layer plays two roles in our 2D FinFETs. Since the gate partially covered the source and drain of devices, HfO_2 layer was firstly used as the insulating spacer to isolate the source/drain and gate electrodes. Simultaneously, HfO_2 layer also acted as dielectric to gain an excellent gate control owing to its reactively high dielectric constant (~ 16).

We fully agree with the referee's comments about the necessity of HfO_2 layer. To make our manuscript more informative to readers, we have included the reason for additional

HfO₂ layer in main text (page 7) as follows: “It worth noting that the native-oxide Bi₂SeO₅ can act as the sole gate dielectric and enable outstanding gate control for the long channel 2D FinFETs (Supplementary Fig. 11). To obtain high-performance 2D FinFETs by shrinking channel length, HfO₂ layer was introduced into the devices. The additional HfO₂ layer acted as an insulating spacer to isolate the source/drain and gate electrodes, and also served as dielectric because of its high dielectric constant ($\kappa \approx 16$).” The new device data without HfO₂ layer has also been added to the revision as Supplementary Figure 11.

Figure Response 12. Electrical performance of 2D Bi₂O₂Se/Bi₂SeO₅ FinFET fabricated without using additional HfO₂ dielectric. **a**, Schematic diagram of 2D Bi₂O₂Se FinFET fabricated with Bi₂SeO₅ dielectric. **b**, **c**, Transfer (b) and output (c) curves measurement results for 50 cycles of a 2D Bi₂O₂Se/Bi₂SeO₅ FinFET with a channel length (L_{ch}) of 3.5 μm . The inset in (b) shows the top-view SEM image of as-fabricated 2D FinFET.

3) *The crystal structures of the native oxide layer on the top edge of the Bi₂O₂Se fins look very different from that on the side surfaces. Please include some description and characterization to compare the two regions and comment on how would such discrepancy may impact the finFET I-V characteristics.*

Authors' response

We thank the referee for raising this concern. During the oxidation, the intercalation will induce lattice expansion on the top edge of the 2D fin-oxide heterostructures, but it does not degrade the electrical properties of as-synthesized fin structures. Therefore, the discrepancy of crystal structures does not impact the FinFET I - V characteristics.

As shown in Figure R13c, when 2D Bi₂O₂Se was oxidized intercalatively into Bi₂SeO₅, lattice strain would be generated at the top interface of 2D fin-oxide heterostructures, owing to the increased c lattice constant from 6.08 \AA to 7.75 \AA . Meanwhile, the

interface of vertical sidewalls is strain-free, for Bi_2SeO_5 and $\text{Bi}_2\text{O}_2\text{Se}$ share the same lattice parameter along $[100]$ and $[010]$ direction (Figure R13d). Since the interfacial quality of vertical sidewalls in 2D $\text{Bi}_2\text{O}_2\text{Se}/\text{Bi}_2\text{SeO}_5$ is predominant for their electrical properties as an ultrathin 2D fin, the interlayer does not impact the transistor characteristics.

Figure Response 13. Interlayer expansion of 2D fin-oxide heterostructure caused by intercalative oxidation from $\text{Bi}_2\text{O}_2\text{Se}$ to Bi_2SeO_5 . **a**, Schematics for the interlayer expansion during intercalative oxidation of 2D $\text{Bi}_2\text{O}_2\text{Se}$ fins. **b**, Cross-sectional STEM image of an 2D $\text{Bi}_2\text{O}_2\text{Se}/\text{Bi}_2\text{SeO}_5$ fin-oxide heterostructure. **c**, Cross-sectional high-resolution STEM image of the interfacial structure on the top of 2D $\text{Bi}_2\text{O}_2\text{Se}/\text{Bi}_2\text{SeO}_5$ fin-oxide heterostructure. **d**, Cross-sectional high-resolution STEM image of the interfacial structure on the vertical sidewalls of 2D $\text{Bi}_2\text{O}_2\text{Se}/\text{Bi}_2\text{SeO}_5$ fin-oxide heterostructure.

4) The authors provided a novel approach to engineer the surface terrains of the growth substrate, which can guide the growth orientation of the $\text{Bi}_2\text{O}_2\text{Se}$ vertical flakes. However, there are still variations in terms of the fin thickness, fin spacing, etc. I was wondering if there is a way to better control the placement and size distributions of the $\text{Bi}_2\text{O}_2\text{Se}$ crystals.

Authors' response

We appreciate the referee for the question. There are indeed variations in terms of the fin thickness and fin spacing in our current fabrication protocol. The fluctuation of fin spacing primarily comes from the variation of step spacing. We have to admit that it is difficult to create an ordered step arrays via a diamond scrape. As shown in Figure R14, after combining a micromachined arm and a diamond scrape, the spacing of 2D fin arrays is controllable by controlling the spacing of step arrays. However, the thickness and spacing of as-grown 2D fins in single arrays is difficult to control. More efforts

have to be made to fabricate 2D fin arrays with more uniform morphology and uniform spacing in the future. For example, nanoimprinting can be employed to fabricate the periodic step arrays, which in turn enables control of the fin spacing finally.

Figure Response 14. 2D fin arrays with controllable fin spacing. **a**, SEM image of guided-grown 2D fin arrays while using a micromachined arm and a diamond scrape to control the step spacing. **b**, Statistical orientations of the 2D fin arrays in (a). Insert: schematic for the fin orientation.

Figure Response 15. Controllability of the fin spacing and thickness of 2D fin-oxide heterostructure arrays. **a**, Top view SEM image of periodic 2D fin arrays with uniform thickness. **b**, **c**, The fin spacing and thickness fluctuation of 2D fin arrays in (a). **d**, Top view SEM image of periodic 2D fin arrays with uniform spacing. **e**, **f**, The fin spacing and thickness fluctuation of 2D fin arrays in (d).

Besides, the fluctuations in fin thickness are the result of kinetic factors, which can be resolved by constituent growth. As shown in Figure R15, we combined maskless laser lithography system and Ar-based ion-beam etching technology to create a periodic nucleation-side arrays. The fin spacing is mainly determined by the step spacing, so as

long as the step spacing is fixed, the fin spacing will fluctuate less. Figure R15b illustrates that the fin spacing of 2D fins is 2.83-2.96 μm . The remained small fluctuation is caused by deviation of step position from step-preparation process. Additionally, the 2D fin arrays exhibits relatively uniform thickness (within a range of 700-800 nm). After adjusting the growth conditions (such as the weight and ratio of precursors), we managed to obtain thinner 2D fin arrays with uniform thickness of 25 nm, but the fluctuation of fin spacing increased slightly (Figure R15d-f).

Combining the above two methodologies, we are fully confident that a more controllable preparation of 2D fin arrays can be realized by using nanoimprinting technique. However, more efforts and time should be required, and we will struggle to achieve the preparation of 2D fin arrays with uniform spacing and thickness in future works based on this method.

5) In the introductory paragraphs, the authors should cite early works for the development of silicon finFET technologies. The authors should also include pervious studies on 3D transistor architectures made with 2D semiconductors (finFETs, nanosheet FETs, and CFETs).

Authors' response

We appreciate the referee's valuable suggestions for citing early works about silicon FinFET and 3D transistor architectures made with 2D semiconductors. We have included relevant literatures in the revision (S. Gupta *et al.*, 7-nm FinFET CMOS Design Enabled by Stress Engineering Using Si, Ge, and Sn, *IEEE Trans. Electron Devices* **2014**, *61*, 1222-1230; C. Lin *et al.*, High Performance 14nm SOI FinFET CMOS Technology with 0.0174 μm^2 embedded DRAM and 15 Levels of Cu Metallization, **2014 IEEE International Electron Devices Meeting (IEDM)**, 3.8.1-3.8.3; X. Huang, P. Zhou *et al.*, Ultrathin Multibrige Channel Transistor Enabled by van der Waals Assembly, *Adv. Mater.* **2021**, *33*, 2102201; L. Tong, P. Zhou *et al.*, Heterogeneous complementary field-effect transistors based on silicon and molybdenum disulfide, *Nat. Electron.* **2022**, *6*, 37-44).

Reviewer #3 (Remarks to the Author):

The authors presented their work on fabricating (multiple) fins primarily made on $\text{Bi}_2\text{O}_2\text{Se}/\text{Bi}_2\text{SeO}_5$ fin-oxide heterostructures, with the fins being grown vertically, through ledge-enhanced epitaxy on LaAlO_3 -terminated surfaces. The authors devote a part of their work to explaining the fabrication process, then explaining the enhancement of the nucleation at the ledge site with support of DFT calculations and, finally, demonstration of finfet devices made with such fin-oxide heterostructures, eventually in a multifin configuration. The devices show good performance, at least in terms of current drive and, taken together, this is a reasonable work, despite several technical inaccuracies, which I will elaborate on later in this review.

Despite all these considerations, I believe the work is not suitable for being published Nature Communications, and I will present below my arguments:

1) When we put this work in context, faced with existing literature, it appears to me that the paper referenced as [15], Tan C. et al, and recently published in Nature (2023), by the same group (first two authors in reversed order), already disclosed most of the findings reported here. Namely: fabrication of the fin, possibility to control their (unidirectional) formation through techniques similar to what is discussed here, and even reported drive currents and on/off current ratio superior to those reported here: i.e. $830\text{uA}/\text{um}$, on/off of $1e7$, etc. When all these are taken into account, the added value of this work is practically reduced to the understanding with the help of DFT. This is, in my opinion, too thin to defend eligibility for Nat. Comm. publication. To be clear: even if there are technical inaccuracies in this work (I will explain), the fundamental issue I have with the publication of this work is the overall novelty and added value for a journal of the caliber of Nat. Comm. I would see no problem in considering this work and improving it further in view of publication in other journals where the emphasis is, for instance, on providing a more in depth understanding and systematic analysis of an already reported novel concept.

Authors' response

We thank the referee for raising this concern. Although both the research in this work and the reported work published in *Nature* are based on 2D fin architectures, we still strongly believe that this work is valuable enough to be published in *Nature Communications*. Below we will detail the differences and highlights.

In this work, driven by ledge-guided epitaxy, we explore the epitaxial growth of mono-oriented 2D fin arrays that are induced to nucleate orientationally due to the preferential adsorption of precursors at the ledges. The orientation of 2D fin seeding relies on the ledges, which can thermodynamically break the dependence on the symmetry of the epitaxial substrate in the oriented nucleation process of 2D fin seeds. Taking the four-

fold symmetric LaAlO₃ (100) as a typical example, 2D Bi₂O₂Se fins grown on the (100)-facet surface generally have two orientations under the random defect-induced nucleation (Tan, C., Peng, H., et al. 2D fin field-effect transistors integrated with epitaxial high-k gate oxide. *Nature* **2023**, 616, 66), yet mono-orientation nucleation can be achieved under ledge-guided nucleation. Differently, the research work published in *Nature* reported a method for epitaxially growing vertical 2D fins on insulating substrates (Tan, C., Peng, H., et al. 2D fin field-effect transistors integrated with epitaxial high-k gate oxide. *Nature* **2023**, 616, 66). In that work, epitaxial growth of mono-oriented 2D fins necessitates coworkers with two-fold symmetric substrates (e.g. MgO (110)) and defect-induced selective nucleation.

On the other hand, to achieve possible applications, it is necessary to explore integration potential of 2D FinFETs. Multi-channel 2D FinFETs fabricated on the guided-grown 2D Bi₂O₂Se/Bi₂SeO₅ fin-oxide heterostructure arrays exhibited excellent performance such as an on/off current ratio greater than 10⁶, high on-state current, low off-state current density, and excellent reliability during repeated measurements and high operating temperature tests. Additionally, our work presented the first demonstration of 2D multi-fin FETs, indicating that 2D multiple-fin arrays can work independently and collaboratively, demonstrating the integration potential of 2D fin arrays. However, the previous work only emphasized the high-performance 2D FinFETs based on single 2D fin. In particular, the electrical reliability and high-operating temperature performances of integrated 2D multi-fin FETs are critical for evaluating device performance yet have never been investigated.

In our revised version, after fully considering the reviewer's concern about the distinctiveness of our work from reported work in *Nature*, we have accordingly revised the sentence in page 2 of main text as follows: "*Remarkably, the ledge-guided epitaxy of mono-oriented 2D fin arrays is independent of the symmetry of substrate, and differs to the recently reported defect-induced epitaxy which necessitates coworkers with two-fold symmetric substrates (e.g. MgO (110)) and defect-induced selective nucleation*".

2) Technical inaccuracies and weaknesses of this work: regardless of my previous point, I have several remarks regarding this work, and outline below a few:

- the fins are thick: 18-24 nm in thickness, that means a multilayer structure. 2D materials are appealing because they are shown to perform at monolayer thickness, a regime that is clearly inaccessible to Si, where severe degradation of the electrical performance onsets at around 5-6nm channel thickness. I think the challenges of this guided (vertical) growth would be huge when trying to approach control of the fin width. Could the authors grow these fins a few monolayers (not to say 1 monolayer) thin? I think this is hitting in fact another key problem connecting to the mechanical stability, which would actually lead to considering fins laid horizontally, not standing vertically (these first ones are also called "nanosheets" in the technologists' slang)

Authors' response

We are grateful to the referee for the insightful comments. Indeed, ultra-thin 2D fins have advantages over Si in electrical performance. Due to the ultrafast growth rate, it is difficult to directly grow 2D Bi₂O₂Se fins with thickness smaller than 5 nm. As shown in Figure R16, the thickness of 2D fins can be tuned by changing the oxygen concentration. With increasing oxygen, the thickness of as-grown 2D Bi₂O₂Se fins was increased. The reason for the above phenomenon is likely related to the oxygen absorption on substrate surface in nucleation process of 2D fins. By controlling the low oxygen concentration and short growth time (~ 10 s), the thickness of 2D fins can reach around 18 nm.

Figure Response 16. Control of the thickness of 2D Bi₂O₂Se fins. **a-c**, SEM images of fins obtained under 0, 20 and 40 ppm O₂, respectively. The growth time is ~10s. **d**, Statistics for fin thickness as a function of oxygen concentration.

Although directly growth of ultra-thin 2D Bi₂O₂Se fins seems challenging, intercalative oxidation is an efficient method to thinning 2D Bi₂O₂Se fins (Figure R17). As the thickness of oxidized Bi₂O₂Se can be well-controlled by time and temperature (Y. Zhang, H. Peng *et al.* A single-crystalline native dielectric for two-dimensional semiconductors with an equivalent oxide thickness below 0.5 nm, *Nature Electron.* **2022**, 5, 643–649), ultra-thin 2D Bi₂O₂Se fins can be obtained after suitable oxidation process. The oxidization process does not destroy the pristine morphology of 2D fins (Figure R17b). As shown in Figure R17c, 2D Bi₂O₂Se fin can be thinned to 3 nm through controlled intercalative oxidation. Such an ultra-thin 2D Bi₂O₂Se fin exhibits excellent mechanical stability.

Figure Response 17. Fabrication of ultra-thin 2D $\text{Bi}_2\text{O}_2\text{Se}$ fin. **a**, Schematic illustration for preparing ultrathin fins by intercalative oxidation. **b**, Tilted-view SEM images of as-synthesized rectangular 2D $\text{Bi}_2\text{O}_2\text{Se}$ fin arrays (left) and $\text{Bi}_2\text{O}_2\text{Se}/\text{Bi}_2\text{SeO}_5$ fin-oxide heterostructure arrays after intercalative oxidation (right) on the MgO (110) surface. **c**, Cross-sectional HR-STEM images of the 2D $\text{Bi}_2\text{O}_2\text{Se}$ fin showing the ultrathin body of about 3 nm.

Remarkably, combining oxidation and selective etching, we can further realize the downscaling of total thickness of 2D fins. For the 2D $\text{Bi}_2\text{SeO}_5/\text{Bi}_2\text{O}_2\text{Se}$ heterostructure, the Bi_2SeO_5 can be selectively etched away via diluted HF acid ($\sim 0.2\%$) to expose fresh and intact $\text{Bi}_2\text{O}_2\text{Se}$ with an atomically smooth surface (T. Li, H. Peng *et al.* A native oxide high- κ gate dielectric for two-dimensional electronics, *Nature Electron.* 2020, 3, 473–478). So, the total thickness of 2D fins can be controllably scaled by oxidation-etching cycle. To determine the surface roughness and thickness evolution by using AFM, the vertical fins were put down on the substrate. As shown in Figure R18a, the thickness of pristine 2D fins can be reduced from ~ 16 nm to ~ 4 nm after two cycles. By adjusting the intercalative oxidation time, the thinning rate of 2D fins can be tuned (Figure R18b). Therefore, the thickness scaling with precise thickness control and atomically smooth surfaces can be achieved by such oxidation-etching cycles.

Figure Response 18. Controllable thickness scaling of 2D fins. To determine the surface roughness and thickness evolution by using AFM, the vertical fins were put down on the substrate. **a**, Thickness scaling of a $\text{Bi}_2\text{O}_2\text{Se}$ fin in two oxidation-etching cycles. **b**, Thickness scaling of $\text{Bi}_2\text{O}_2\text{Se}$ fins with different oxidation rate.

In our revised version, Figure R17 and Figure R18 have been included as supplementary Fig. 8 and supplementary Fig. 10. Accordingly, we have revised “*On the one hand, controllable oxidation is an effective approach to thinning 2D fins in the fin-oxide heterostructures, which enables 2D fin as thin as 3 nm (Supplementary Fig. 8e). In addition, combining oxidation and selective etching via diluted HF acid (~0.2 %), the downscaling of total thickness of 2D fins can possibly be further realized (Supplementary Fig. 10)*” in the main text, page 6.

- the authors use repeatedly the term "integration". I do not see much integration here; there is a long, long way from these structures to integrated devices in the common acceptance of the term, used in the semiconductor tech industry and R&D.

Authors’ response

we are grateful for the referee’s concerns. In our work, the term “integration” is used with two main considerations. With respect to the architecture of devices, the term “integration” is adopted to reveal the epitaxial process of combing high-mobility 2D semiconductor fin $\text{Bi}_2\text{O}_2\text{Se}$ with single-crystal high- κ gate oxide Bi_2SeO_5 in a critical structure for 2D FinFETs. Indeed, the term “integration” is widely adopted to describe the integration of semiconductor channels and dielectric layers of transistor (Liu, Y., et al. Promises and prospects of two-dimensional transistors, *Nature* **2021**, 591, 43; Huang, J.-K., et al. High- κ perovskite membranes as insulators for two-dimensional transistors. *Nature* **2022**, 605, 262; Yang, X., et al. Highly reproducible van der Waals integration of two-dimensional electronics on the wafer scale. *Nat. Nanotechnol.* **2023**, 18, 471). Besides, the term “integration” is employed to illustrate that the multiple fin channels was fused into a single unite to fabricate multi-fin FETs, and improving the driving

capability.

In our revised manuscript, to avoid possible misunderstanding, we have modified the relevant expressions to be more accurate, such as “*High-density aligned 2D fin arrays facilitate the fabrication of 2D multi-fin FETs that integrate multiple fin channels, which have the potential to boost electrical performance with higher drive capability*” and “*Notably, the fin-oxide heterostructure with atomically smooth interface, formed by the 2D Bi₂O₂Se fin integrated high- κ native-oxide Bi₂SeO₅ epilayer, acts as the workhorse architecture for 2D multi-fin FETs (Supplementary Fig. 8d)*” in page 6.

- The authors report that their fin arrays could "meet the requirement of advanced 3-nm technology node, as projected by the International Roadmap for Devices and Systems" (p.5, 190-192). As a matter of fact, the 3-nm node is in production this year, and not by one manufacturer, but 2, and another one on its way. Then, why would it be needed to do research with (still) exotic materials for something that's available commercially? If IRDS is brought into the picture, I would expect to focus on the nodes that are still in R&D, and have not yet solutions available in all areas, and where 2D materials are expected to make a difference. These would be 1nm eq. and beyond, with a time horizon for reaching industrial maturity in the next decade.

Authors' response

We are very thankful for the reviewer's constructive suggestions.

As the predominant transistor architecture, the FinFETs fabricated on the Si fin-shaped channels have driven the device downscaling to 3-nm technology node in integrated circuits. However, for the forthcoming sub-1-nm nodes at the ultimate scaling limits, the physical limits of sub-5-nm Si body thickness and imperfect surface/interfaces scattering impede the carrier mobility and drive current, degrading the device performance. In this regard, 2D layered semiconductors have the capability of scaling FETs to the 1-nm node and beyond because of their intrinsic atom-scale thicknesses (K. Banerjee *et al.* The future transistors, *Nature* **2023**, 620, 501). The latest International Roadmap of Devices and Systems (IRDS) shared across chip manufacturers, material suppliers, and apparatus makers has projected that the channel material evolving from Si to 2D semiconductors (More Moore table. *International Roadmap for Devices and Systems IRDS 2022 More Moore (ieee.org) (2022)*). Additionally, 2D-fin-based device architectures (FinFETs and VGAA) enable excellent gate controllability, high driving current and integration density, which are being aggressively explored as alternative technologies for the ultra-scaled transistors to fulfill the stringent requirements of sub-1-nm technology node (K. Banerjee *et al.* The future transistors, *Nature* **2023**, 620, 501; Liu, Y., et al. Promises and prospects of two-dimensional transistors, *Nature* **2021**, 591, 43). Therefore, upon the aforementioned factors, epitaxial high-density 2D fin arrays hold possibility to support material and architectural platforms for building ultra-scaled

transistors that meet the requirements of advanced sub-1-nm technology nodes.

We fully agree with the review's concern that the channel material requirements of 3-nm technology node could be misinterpreted. Therefore, to avoid possible misunderstandings, we have adopted a more accurate expression as follows: "*Given more precise and controllable preparation of substrate steps, it will be possible to fabricate ordered high-density 2D fin arrays that meet the material requirement of advanced sub-1-nm technology node*".

REVIEWER COMMENTS

Reviewer #1 (Remarks to the Author):

The authors have addressed my questions, particularly concerning the growth mechanism. Yet, the explanation for these energetically preferred steps' large-scale and selective formation at an "atomic" resolution prompts further inquiry. It's intriguing how a mere scratch from a diamond tip can induce self-aligned step edges that are perfectly orthogonal to the scratch direction, not to mention the ability to dictate terrace width and step height. Moreover, the influence of varying step heights on the aspect ratios of the resultant 2D Bi₂O₂Se fin arrays warrants clarification. Could steeper steps lead to thicker fins, for example? It would be beneficial to include a series of cross-sectional HRTEM images capturing the nucleation stages at step edges of differing heights to shed light on this. Regarding device performance, comparing the best-in-class 2D TMD transistors would be more appropriate than contrasting with 2D TMD "FinFETs," especially as we approach the frontier of the 1 nm-node era, surpassing silicon-based FETs.

Reviewer #2 (Remarks to the Author):

In the revision, the authors have made tremendous efforts in addressing the reviewers' comments. The revised manuscript is in a good shape and ready for publication. I mentioned in the previous referee's report that the authors should cite the early works on Si FinFETs and previous studies on 2D semiconductor finFETs, nanosheetFETs and CFETs. Some of the cornerstone studies are still missing. Below are a few examples:

Hisamoto, Digh, et al. "A folded-channel MOSFET for deep-sub-tenth micron era." IEDM Tech. Dig 1998 (1998): 1032-1034.

Hisamoto, Digh, et al. "FinFET-a self-aligned double-gate MOSFET scalable to 20 nm." IEEE transactions on electron devices 47.12 (2000): 2320-2325.

Huang, Xuejue, et al. "Sub 50-nm finfet: Pmos." International Electron Devices Meeting 1999. Technical Digest (Cat. No. 99CH36318). IEEE, 1999.

Yu, Bin, et al. "FinFET scaling to 10 nm gate length." Digest. International Electron Devices Meeting,. IEEE, 2002.

Lee, Hyunjin, et al. "Sub-5nm all-around gate FinFET for ultimate scaling." 2006 Symposium on VLSI Technology, 2006. Digest of Technical Papers.. IEEE, 2006.

APA

Xiong, Xiong, et al. "Demonstration of Vertically-stacked CVD Monolayer Channels: MoS₂ Nanosheets GAA-FET with $I_{on} > 700 \mu\text{A}/\mu\text{m}$ and MoS₂/WSe₂ CFET." 2021 IEEE International Electron Devices Meeting (IEDM). IEEE, 2021.

Chung, Yun-Yan, et al. "First Demonstration of GAA Monolayer-MoS₂ Nanosheet nFET with $410 \mu\text{A}/\mu\text{m}$ I_{D1V} at 40nm gate length." 2022 International Electron Devices Meeting (IEDM). IEEE, 2022.

Reviewer #3 (Remarks to the Author):

The authors paid attention to the concerns I raised in my initial review.

I agree with this work being valuable and have publication potential, however not for Nature Comm. for (once again) two reasons: first, despite having differentiating elements, I believe it is not novel/distinct enough from previous works of the group and for (given the context set by the first reason) not demonstrating beyond-incremental improvement in relation to current literature/SoA. (in short - it should be clearly novel or better or, ideally, both).

I accept the authors "modulation" of the integration claim, in the meaning provided through their newly introduced sentence.

I do not agree with the statement that thinning the thickness by oxidation is a viable solution. First, the limit of the fin still remains 3-4nm, which might be different from what is targeted by 2D materials and second, while the fin thickness may eventually be reduced to 4nm starting from 16nm, the pitch will not reduce at all. That means there is no gain in density for a fin of 4nm, compared to the one of 16nm, since the distance between 2 neighboring fins will increase (by this trimming) with $2 \times 12\text{nm} = 24\text{nm}$ (to which we would need to add the initial separation between the "thick" original fins).

I believe the authors are overoptimistic with their last claim, regarding the possibility to "fabricate ordered high-density 2D fin arrays that meet the material requirement of advanced sub-1-nm technology node" using this approach. As a matter of fact (beside the claim of "high density", which is questionable - see above paragraph), it has taken more than a decade to bring "conventional" 2D TMD materials to the fab, and it might take another one to see them getting out, packaged in chips (and this is more or less where the 1nm. node is).

REVIEWER COMMENTS

Reviewer #1 (Remarks to the Author):

The authors have addressed my questions, particularly concerning the growth mechanism.

Authors' response

We really appreciate the reviewer's constructive suggestions on our work. The suggestions from the referee help bring significant improvements to our manuscript. Our point-by-point responses are listed below.

1) Yet, the explanation for these energetically preferred steps' large-scale and selective formation at an "atomic" resolution prompts further inquiry. It's intriguing how a mere scratch from a diamond tip can induce self-aligned step edges that are perfectly orthogonal to the scratch direction, not to mention the ability to dictate terrace width and step height.

Authors' response

We are very thankful for the referee raising this concern about the formation of artificially aligned steps. As shown in Figure R1, the self-aligned step edges are artificial created by sliding a less-sharp diamond tip passing through the substrate. To ensure that the formed edges orthogonal to the scratch direction, the sliding direction must parallel or perpendicular to the [001] or [010] direction of the LaAlO_3 (100) substrate. Accordingly, the step edges formed with an "atomic" resolution originate from brittle fractures along the [001] or [010] direction of the LaAlO_3 lattice from the [010] or [001] cleavage plane. From the experimental epitaxial results, despite the fact that the formed step edges are not perpendicular to the scratches, a mono-oriented 2D fin array is still obtained (Figure R1e, f).

We have to admit that it is not easy to accurately control step width and step height with a diamond scraper. In this work, we demonstrate mechanistically that the preparation of mono-oriented 2D fin arrays can be easily achieved by using this mechanical scratching approach. We are confident that the steps can be more precise controlled by using more controllable nanofabrication techniques, such as nanoimprinting.

We have revised "taking the adopted LaAlO_3 (100) as a presentative example, several artificially self-aligned steps with specific orientation on the LaAlO_3 (100) surface can be easily pre-created..." in the main text.

Figure Response 1. The relationship between scratch direction and substrate step orientation. **a**, Schematic illustration of creating line-array scratches by diamond scraper. **b, c**, AFM image (**b**) and corresponding step height (**c**) of the artificial steps. **d**, Schematic illustration of different step orientations under different scratch directions. **e**, SEM images of 2D fin arrays with varying orientations resulting from scratches in different directions.

2) Moreover, the influence of varying step heights on the aspect ratios of the resultant 2D $\text{Bi}_2\text{O}_2\text{Se}$ fin arrays warrants clarification. Could steeper steps lead to thicker fins, for example? It would be beneficial to include a series of cross-sectional HRTEM images capturing the nucleation stages at step edges of differing heights to shed light on this.

Authors' response

We are very thankful for the comment raised by the referee. The artificial steps generated through the diamond scraper have a height of approximately 10 nm, consisting of numerous atomic steps. Within this range, we find no substantial correlation between step height and fin thickness. On the other hand, we concur with

the reviewer about capturing the nucleation stages through a series of cross-sectional HRTEM. However, due to the ultrafast growth rate of 2D fins, maintaining the fins at the micronucleus stage proves challenging. We will conduct an in-depth and systematic study on the nucleation of the 2D fins in the future, probably by using in situ techniques.

3) Regarding device performance, comparing the best-in-class 2D TMD transistors would be more appropriate than contrasting with 2D TMD "FinFETs," especially as we approach the frontier of the 1 nm-node era, surpassing silicon-based FETs.

Authors' response

We are very grateful to the reviewer's constructive suggestion. As shown in Figure R2, we have compared the device performance of 2D Bi₂O₂Se FinFET and that of the best-in-class 2D TMD transistors and included it in Figure 4 of the revised manuscript.

Figure Response 2. Comparison of normalized current of the fabricated 2D multi-fin FETs with 2D MoS₂ FET, 2D InSe FET, 2D MoS₂ FinFETs and Intel's 14nm-node Si FinFETs under low gate-voltage modulation.

Reviewer #2 (Remarks to the Author):

In the revision, the authors have made tremendous efforts in addressing the reviewers' comments. The revised manuscript is in a good shape and ready for publication. I mentioned in the previous referee's report that the authors should cite the early works on Si FinFETs and previous studies on 2D semiconductor finFETs, nanosheetFETs and CFETs. Some of the cornerstone studies are still missing. Below are a few examples:

Hisamoto, Digh, et al. "A folded-channel MOSFET for deep-sub-tenth micron era." IEDM Tech. Dig 1998 (1998): 1032-1034.

Hisamoto, Digh, et al. "FinFET-a self-aligned double-gate MOSFET scalable to 20 nm." *IEEE transactions on electron devices* 47.12 (2000): 2320-2325.

Huang, Xuejue, et al. "Sub 50-nm finfet: Pmos." *International Electron Devices Meeting 1999. Technical Digest (Cat. No. 99CH36318)*. IEEE, 1999.

Yu, Bin, et al. "FinFET scaling to 10 nm gate length." *Digest. International Electron Devices Meeting.*. IEEE, 2002.

Lee, Hyunjin, et al. "Sub-5nm all-around gate FinFET for ultimate scaling." *2006 Symposium on VLSI Technology, 2006. Digest of Technical Papers*. IEEE, 2006.

Xiong, Xiong, et al. "Demonstration of Vertically-stacked CVD Monolayer Channels: MoS₂ Nanosheets GAA-FET with Ion > 700 $\mu\text{A}/\mu\text{m}$ and MoS₂/WSe₂ CFET." *2021 IEEE International Electron Devices Meeting (IEDM)*. IEEE, 2021.

Chung, Yun-Yan, et al. "First Demonstration of GAA Monolayer-MoS₂ Nanosheet nFET with 410 $\mu\text{A}/\mu\text{m}$ ID at 1V VD at 40nm gate length." *2022 International Electron Devices Meeting (IEDM)*. IEEE, 2022.

Authors' response

We appreciate the referee's positive comments on our revised manuscript. We have included the missing cornerstone studies about silicon FinFET and 3D transistor architectures made with 2D semiconductors in the new revision (Refs. 10-14, 26-27).

10 Hisamoto, D. *et al.* A folded-channel MOSFET for deep-sub-tenth micron era. 1998 IEEE International Electron Devices Meeting (IEDM), 1032-1034 (IEEE, 1998).

11 Hisamoto, D. *et al.* FinFET-a self-aligned double-gate MOSFET scalable to 20 nm. *IEEE transactions on electron devices* 47, 2320-2325 (2000).

12 Huang, X. *et al.* Sub 50-nm FinFET: PMOS. 1999 IEEE International Electron Devices Meeting (IEDM), 67-70 (IEEE, 1999).

13 Yu, B. *et al.* FinFET scaling to 10 nm gate length. Digest. 2002 IEEE International Electron Devices Meeting (IEDM), 251-254 (IEEE, 2002).

14 Lee, H. *et al.* Sub-5nm all-around gate FinFET for ultimate scaling. 2006 Symposium on VLSI Technology (VLSIT), 58-59 (IEEE, 2006).

26 Xiong, X. *et al.* Demonstration of Vertically-stacked CVD Monolayer Channels: MoS₂ Nanosheets GAA-FET with Ion > 700 $\mu\text{A}/\mu\text{m}$ and MoS₂/WSe₂ CFET. 2021 IEEE International Electron Devices Meeting (IEDM), 7.5.1-7.5.4 (IEEE, 2021).

27 Chung, Y. *et al.* First Demonstration of GAA Monolayer-MoS₂ Nanosheet nFET with 410 $\mu\text{A}/\mu\text{m}$ ID at 1V VD at 40 nm gate length. 2022 International Electron Devices Meeting (IEDM), 34.5.1-34.5.4 (IEEE, 2022).

Reviewer #3 (Remarks to the Author):

The authors paid attention to the concerns I raised in my initial review.

I agree with this work being valuable and have publication potential, however not for Nature Comm. for (once again) two reasons: first, despite having differentiating elements, I believe it is not novel/distinct enough from previous works of the group and for (given the context set by the first reason) not demonstrating beyond-incremental improvement in relation to current literature/SoA.

(in short - it should be clearly novel or better or, ideally, both).

I accept the authors "modulation" of the integration claim, in the meaning provided through their newly introduced sentence.

Authors' response

We appreciate the referee's comments on the improved quality of our revised manuscript but we respectfully disagree that our work is *not novel/distinct enough from previous works of the group* (Nature 2023, 616, 66). Following other reviewers' and editorial standpoint "the reported multi-fin FET integration noteworthy and sufficiently distinct from previous work (Nature 2023, 616, 66)", we believe our work provides an important method to achieve high-density mono-oriented 2D fin arrays, which can potentially be the central material system for the future ultra-scaled transistors. By carefully reflecting on the comments, our point-by-point responses are listed below.

*I do not agree with the statement that thinning the thickness by oxidation is a viable solution. First, the limit of the fin still remains 3-4nm, which might be different from what is targeted by 2D materials and second, while the fin thickness may eventually be reduced to 4nm starting from 16nm, the pitch will not reduce at all. That means there is no gain in density for a fin of 4nm, compared to the one of 16nm, since the distance between 2 neighboring fins will increase (by this trimming) with $2*12nm=24nm$ (to which we would need to add the initial separation between the "thick" original fins).*

Authors' response

We are grateful for the referee's comments. As suggested by the International Roadmap for Devices and Systems (IRDS) in 2023, the device requires a minimum channel thickness of 4 nm at the 5 Å technology node (<https://irds.ieee.org/> (2023)). In our work, the thickness of 2D fin can be reduced to 3 nm, which perfectly matches the channel thickness requirements of the state-of-the-art nodes forecasted by 2023 IRDS roadmap.

As for the distance between two neighboring fins, we recognize that more time are needed to reduce the spacing of the 2D fins by means of vapour-phase epitaxy. To explore an effective strategy to further reduce the distance of two neighboring fins, we employed an ion-beam etching (IBE) approach to etch the as-synthesized fins by using argon gas as a medium. As shown in Figure R3, the etched fins form a hollow square-hole and square-hole 2D fin still maintain vertically freestanding. Remarkably, the distance between two neighboring fins in a synthesized square-hole 2D fin can be successfully reduced to 10 nm. This effective approach will be elaborated detailed in our future work, not in this work.

Figure Response 3. IBE etching process for fabricating square-hole 2D fin with small spacing. **a**, Schematic illustration for the fabrication process of square-hole 2D fin via ion-beam etching. **b**, *in-situ* SEM characterization of 2D fin before and after etching by IBE. **c**, **d**, SEM images of square-hole 2D fins with distance as small as 10 nm between two neighboring fins.

I believe the authors are overoptimistic with their last claim, regarding the possibility to "fabricate ordered high-density 2D fin arrays that meet the material requirement of advanced sub-1-nm technology node" using this approach. As a matter of fact (beside the claim of "high density", which is questionable - see above paragraph), it has taken more than a decade to bring "conventional" 2D TMD materials to the fab, and it might take another one to see them getting out, packaged in chips (and this is more or less where the 1nm.eq. node is).

Authors' response

We are thankful for the referee's concerns. According to the latest International Roadmap for Devices and Systems (IRDS) projections, transistors fabricated on two-dimensional (2D) semiconductors are expected to realize the ultimate power scaling at

the foreseeable 0.7 nm (sub-1-nm) technology node. Particularly, the 2D fins hold potential to fabricate vertical gate-all-around (VGAA) transistors to enable the relaxation of the length scaling, and an area and cost reduction without a leakage penalty, further improving device performance, energy efficiency and integration density (Cao, W., *et al.* "The future transistors" *Nature* 2023, 620, 501-515). Therefore, it is reasonable to conclude that the fabrication of ordered, high-density 2D fin arrays meet the material requirements of advanced sub-1-nm technology nodes, although there are many challenges in progressing 2D FinFETs from concept genesis to industrial maturity. For Lab-to-Fab transition of 2D transistors, robust synthesis of wafer-scale, high uniformity and reproducibility 2D single crystals or periodic arrays of single crystals at predesignated locations, as well as industry-compatible integration process are essential but challenging. Even though challenges remain, there seems to be no fundamental roadblock to the wafer-scale processing of 2D transistors and the industrial attempt for their production [Liu, Y., *et al.* "Promises and prospects of two-dimensional transistors". *Nature* 2021, 591, 43–53 (perspectives)]. By further optimizing the formation of high-density aligned surface steps with precise spacing control, it is possible to achieve higher-density 2D fin arrays. We believe this advancement will facilitate large-scale integration of 2D multi-fin FETs, thus allowing for further 2D transistor scaling.

REVIEWERS' COMMENTS

Reviewer #1 (Remarks to the Author):

The authors have successfully addressed the bulk of my concerns, particularly regarding the structural analyses. The inclusion of comprehensive fabrication details marks a pivotal advancement, effectively narrowing the divide between laboratory-scale fabrication and industrial manufacturing. This represents a significant leap towards the integration of 2D materials into the realm of next-generation electronic devices. However, we all know that 2D Fins FETs still have a long way to go since there is room for improvement in density, control of fin thickness, and selective growth.